



# Predicting the soil water retention curve from the particle size distribution based on a pore space geometry containing slit-shaped spaces

**Chen-chao Chang**[TS1][1,2] **and Dong-hui Cheng**[1,2]

[1]School of Environmental Sciences and Engineering, Chang'an University, Xi'an, 710054, China
[2]Key Laboratory of Subsurface Hydrology and Ecological Effects in Arid Region, Chang'an University, Ministry of Education, Xi'an, China

**Correspondence:** Dong-hui Cheng (chdhbsh@chd.edu.cn)

**Abstract.** Traditional models employed to predict the soil water retention curve (SWRC) from the particle size distribution (PSD) always underestimate the water content in the dry range of the SWRC. Using the measured physical parameters of 48 soil samples from the UNSODA unsaturated soil hydraulic property database, these errors were proven to originate from an inaccurate estimation of the pore size distribution. A method was therefore proposed to improve the estimation of the water content at high suction heads using a pore model comprising a circle-shaped central pore connected to slit-shaped spaces. In this model, the pore volume fraction of the minimum pore diameter range and the corresponding water content were accordingly increased. The predicted SWRCs using the improved method reasonably approximated the measured SWRCs, which were more accurate than those obtained using the traditional method and the scaling approach in the dry range of the SWRC.

## 1 Introduction

The soil water retention curve (SWRC), which represents the relationship between the water pressure and water content, is fundamental to researching water flow and chemical transport in unsaturated media (Pollacco et al., 2017). Direct measurements of the SWRC consume both time and money (Arya and Paris, 1981; Mohammadi and Vanclooster, 2011), while estimating the SWRC from the particle size distribution (PSD) is both rapid and economical. Therefore, a number of associated conceptual and physical models have been proposed.

The first attempt to directly translate a PSD into an SWRC was made by Arya and Paris (1981) (hereafter referred to as the AP model). In this model, the PSD is divided into multiple size fractions and the bulk and particle densities of the natural-structure samples are uniformly applied to each particle size fraction, from which it follows that the pore fraction and the corresponding solid fraction are equal. Thus, the degree of saturation can be set equal to the cumulative PSD function. The soil suction head can be obtained using the capillary equation based on a "bundle of cylindrical tubes" model, and the pore size in the equation is determined by scaling the pore length and pore volume (Arya et al., 2008). Based on the principle of the AP model, many researchers have focused on improving the suction head calculations, which are commonly based on the capillary equation; but methods that are used to translate the particle diameter into the pore diameter are different (Haverkamp and Parlange, 1986; Zhuang et al., 2001; Mohammadi and Vanclooster, 2011; Jensen et al., 2015). Some models estimate the pore diameter based on the particle packing patterns (e.g., the MV model) (Meskini-Vishkaee et al., 2014), while others utilize the proportionality factor between the pore size and the associated particle diameter (e.g., the HP model and the two-stage approach) (Haverkamp and Parlange, 1986; Jensen et al., 2015). However, the scheme employed to estimate the water content has not been modified and follows the approach of the AP model. The SWRC prediction models which use the same scheme to predict the water content and

Please note the remarks at the end of the manuscript.

only improve the suction head calculation are referred to as the traditional models in the following text.

However, these traditional models underestimate the water content in the dry range of the SWRC (Hwang and Powers, 2003 TS2; Meskini-Vishkaee et al., 2014). Therefore, some researchers have attempted to improve the water content calculation by attributing model errors to both a simplified pore geometry and an incomplete desorption of residual water in the soil pores within a high suction head range (Tuller et al., 1999; Mohammadi and Meskini-Vishkaee, 2012). Recent findings have revealed the existence of corner water, lens water and water films in soils at high matric suction heads (Tuller et al., 1999; Mohammadi and Meskini-Vishkaee, 2012; Or and Tuller, 1999; Shahraeeni and Or, 2010; Tuller and Or, 2005). Therefore, Mohammadi and Meskini-Vishkaee (2012) predicted an SWRC based on the PSD while considering the adsorbed water films and lens water between the soil particles, and slightly improved upon the traditional MV model. Tuller et al. (1999) proposed a pore space geometry containing slit-shaped spaces and derived a corresponding SWRC that considered both the water films and water inside the angular-shaped pores; however, the predicted SWRC failed to describe experimental data at an intermediate water content due to the limitations of the gamma distribution function used to characterize the pore size distribution (PoSD) (Lebeau and Konrad, 2010). Moreover, this model was mathematically complex. Mohammadi and Meskini-Vishkaee (2013) incorporated the residual water content into the MV model and consequently decreased the magnitude of the underestimation in the dry range of the SWRC. However, an accurate estimation of the residual water content remains a challenge. Meskini-Vishkaee et al. (2014) improved the traditional MV model by defining a soil particle packing scaling factor. This method could improve the estimation of the SWRC, and is particularly significant for fine- and medium-textured soils.

Many traditional models are based on a "bundle of cylindrical tubes" representation of the pore space geometry (Arya and Paris, 1981; Zhuang et al., 2001), which results in intrinsic errors when predicting the water flow in variably saturated soils. Consequently, some researchers have considered pore networks as bundles of triangular tubes, which could incorporate the contribution of water in pore corners to the water content (Helland and Skjæveland, 2007). A new pore geometry model comprised of a polygon-shaped central pore connected to slit-shaped spaces was proposed by Tuller et al. (1999) to provide a more realistic representation of natural pore spaces (Tuller et al., 1999; Or and Tuller, 1999; Tuller and Or, 2001). This pore model could represent a foundation for accurately describing the water status in natural soils, particularly in arid environments.

Therefore, the objectives of this study were to evaluate the leading factors that lead to an underestimation of the water content in the dry range of the predicted SWRC using traditional methods and to furthermore propose a method for accurately estimating the water content using a pore space geometry containing slit-shaped spaces to improve the prediction of the SWRC.

## 2 Basic descriptions

The relationship between the PSD and the PoSD is a fundamental element when predicting the SWRC from the PSD. Hwang and Powers (2003) found that the nonlinear relationship between the PSD and the PoSD is more appropriate than the linear relationship applied in the AP model and therefore described both the PSD and the PoSD as lognormal distributions. However, since the PSD and PoSD of soils do not strongly follow a lognormal distribution, this model performed very poorly for moderately fine-textured soils (Hwang and Choi, 2006). Obtaining an accurate PoSD from the PSD of a soil is highly difficult, and the errors that arise from this approach could cause inevitable errors in the predicted SWRC. However, the underestimation of the water content in the dry range of an SWRC has not been comprehensively evaluated from this perspective.

In this study, the measured PoSDs of 48 soil samples were compared with the PoSDs calculated using a traditional model (they were actually the corresponding PSDs) to identify the origins of the errors and their effects on the accuracy of the predicted SWRC. The provided 48 soil samples exhibited a wide range of physical properties (Table 1) and were selected from the UNSODA unsaturated soil hydraulic property database, which contains 790 soil samples with general unsaturated soil hydraulic properties and basic soil properties (e.g., water retention, hydraulic conductivity, soil water diffusivity, PSD, bulk density, and organic matter content) (Nemes et al., 2001). The maximum, minimum and mean values of the soil bulk density and the percentages of clay and sand of the used soil samples for the calibration stage are presented in Table 2.

1. Calculating the PoSD using a traditional model

   Traditional models commonly assume that the pore volume fraction of each size fraction can be set equal to the corresponding solid fraction (Arya and Paris, 1981). Thus, the cumulative pore volume fraction can take the following form:

$$\sum_{j=1}^{j=i} v_j = \sum_{j=1}^{j=i} \omega_j; \ i = 1, 2, \dots n, \tag{1}$$

   where TS4 $\omega_j$ is the solid fraction of the $j$th particle fraction, $v_j$ is the pore volume fraction associated with the $j$th fraction, and $n$ is the total number of size fractions in the PSD.

   The routine procedures employed among the several traditional models to translate a particle diameter into a

**Table 1.** Codes and texture classes of the 48 soils selected from UNSODA.

| UNSODA codes | Texture class |
|---|---|
| 4681, 4680, 2362, 2360, 1400, 1383, 4121, 1361, 2340 | Clay |
| 3191, 1091, 2530, 2531 | Loam |
| 2102, 3150, 3161, 3171, 1160, 3170, 3130, 1031, 4011, 4020 | Loamy sand |
| 1464, 1466, 2100, 3340, 4650, 3142, 1050, 1023, 3141, 3163, | Sand |
| 3164, 3165, 3172, 4051, 4520, 4521 | |
| 3202 | Sandy clay loam |
| 3200, 3203, 4162 | Sandy loam |
| 4042, 4180, 4070, 4673, 1341 | Silt loam |

**Table 2.** Basic soil properties of 48 samples for the model calibration.

| Soil texture | Number of soil samples | | Clay (%) | Sand (%) | $\rho_b$ (g cm$^{-3}$) |
|---|---|---|---|---|---|
| | | Min | 41.5 | 6.1 | 1.08 |
| Clay | 9 | Max | 58.2 | 36.0 | 1.64 |
| | | Average | 50.2 | 14.1 | 1.29 |
| | | Min | 14.0 | 42.0 | 1.36 |
| Loam | 4 | Max | 23.0 | 67.0 | 1.63 |
| | | Average | 17.3 | 50.5 | 1.46 |
| | | Min | 3.0 | 76.2 | 1.32 |
| Loamy sand | 10 | Max | 10.4 | 89.4 | 1.60 |
| | | Average | 6.1 | 83.2 | 1.46 |
| | | Min | 0.7 | 89.6 | 1.41 |
| Sand | 16 | Max | 4.6 | 98.9 | 1.70 |
| | | Average | 2.5 | 93.4 | 1.55 |
| Sandy clay loam | 1 | | 2.7 | 62.5 | 1.70 |
| | | Min | 10.5 | 64.9 | 1.27 |
| Sandy loam | 3 | Max | 19.4 | 76.3 | 1.70 |
| | | Average | 15.0 | 68.8 | 1.50 |
| | | Min | 10.5 | 21.0 | 1.49 |
| Silt loam | 5 | Max | 15.7 | 34.8 | 1.56 |
| | | Average | 12.6 | 26.5 | 1.52 |

pore diameter are different. The equivalent pore diameter can be derived from physical properties, including the bulk density and the particle density, or from the proportionate relationship between the pore size and associated particle diameter. Although the former can logically characterize a pore, a complicated pattern can slightly reduce the model performance, while the latter approach is easy to use, and its rationality has been demonstrated by some researchers (Hamamoto et al., 2011; Sakaki et al., 2014). Here, the latter technique is applied, and it can be expressed as

$$d_i = 0.3 D_i, \qquad (2)$$

where $D_i$ is the mean particle diameter of the $i$th fraction (µm) and $d_i$ is the corresponding equivalent pore

diameter (µm). Inputting the PSD data, the calculated pore diameters are sequentially paired with corresponding pore volume fractions to obtain a calculated PoSD.

2. Estimating the PoSD from the measured SWRC

It is generally difficult to measure the PoSD of a soil; however, the PoSD can be indirectly obtained using the measured water content and suction head (Jayakody et al., 2014). The cumulative pore volume fraction of the $i$th fraction is equal to the ratio of the measured water content to the saturated water content (Eq. 3):

$$\theta_i = \theta_s \sum_{j=1}^{j=i} v_j; \quad i = 1, 2, \ldots n, \qquad (3)$$

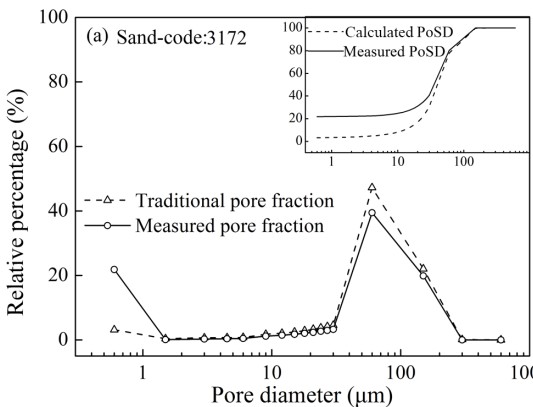 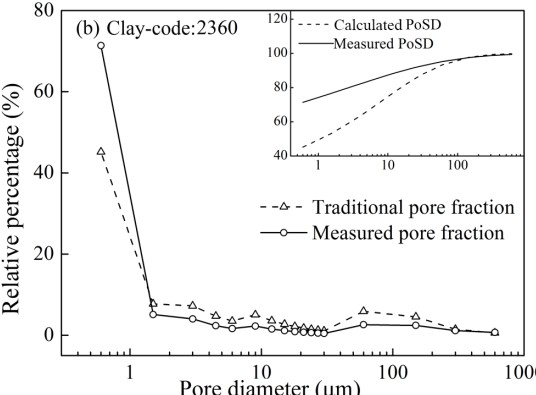

**Figure 1.** Measured vs. calculated pore volume fraction curves using the traditional method TS3 for **(a)** sand (code: 3172) and **(b)** clay (code: 2360). The measured and calculated PoSDs are in the insets at the top of the figures.

where $\theta_s$ is the saturated water content ($\text{cm}^3\,\text{cm}^{-3}$) and $\theta_i$ is the measured water content ($\text{cm}^3\,\text{cm}^{-3}$).

Meanwhile, the corresponding pore diameters are derived on the basis of Laplace's equation and Eq. (4) TS5:

$$\psi_i = \frac{2\sigma\cos\varepsilon}{r_i\,g\,\rho_w}, \qquad (4)$$

where $\psi_i$ is the suction head (mH$_2$O TS6), $\sigma$ is the surface tension ($\text{kg}\,\text{s}^{-2}$), $\varepsilon$ is the contact angle between the soil particle and water, $r_i$ is the pore radius (m), and $\rho_w$ is the density of water ($\text{kg}\,\text{m}^{-3}$). Assuming for water at 20 °C $\sigma = 7.275 \times 10^{-2}\,\text{kg}\,\text{s}^{-2}$, $\rho_w = 998.9\,\text{kg}\,\text{m}^{-3}$, $g = 9.81\,\text{m}\,\text{s}^{-2}$, and $\varepsilon = 0°$ (Mohammadi and Vanclooster, 2011), then transforming $r_i$ to $d_i$ and substituting numerically the values of the constant yield a simplified expression as Eq. (5):

$$\psi_i = \frac{3000}{d_i}, \qquad (5)$$

where $\psi_i$ is the suction head (cmH$_2$O) and $d_i$ is the pore diameter (µm). Then, the pore diameters calculated by Eq. (5) were sequentially paired with the cumulative pore volume fractions calculated by Eq. (3) to obtain a PoSD, which could be considered a measured PoSD.

The calculated and measured PoSD data were fitted using a modified logistic growth model (Eq. 6) (Liu et al., 2003):

$$w_i = \frac{1}{1 + a\exp\left(-bd_i^c\right)}, \qquad (6)$$

where $w_i$ is the cumulative pore volume fraction with diameters smaller than $d_i$ (%) and $a$, $b$, and $c$ are the fitting parameters (dimensionless). This model produced a good fit for the PoSD data employed in this study with a coefficient of determination ($r^2$) that ranged from 0.972 to 0.999.

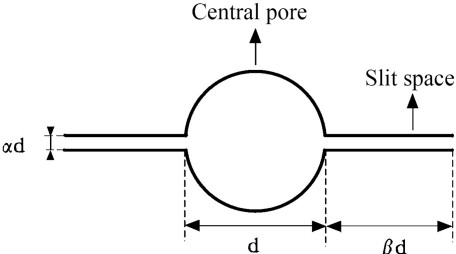

**Figure 2.** Pore model containing two slit-shaped spaces ($d$ denotes the diameter of the central pore, and $\alpha d$ and $\beta d$ denote the widths and lengths of the slit-shaped spaces, respectively).

The measured pore volume fraction curves for the typical samples, namely, sand (code: 3172) and clay (code: 2360), and their calculated curves using the traditional model are presented in Fig. 1. The small maps embedded in Fig. 1 exhibit the measured and calculated PoSD curves, which show that the calculated PoSD curves approximately coincide with the measured curves in the larger pore diameter range, while in the smaller range, which corresponds to the higher suction head range on the SWRC, the calculated values are obviously smaller than the measured values. The underestimation of the pore volume fraction in the smaller pore diameter range can consequently lead to an underestimation of the water content at high suction heads. In particular, the calculated pore volume fraction associated with the smallest pore diameter ($d \leq 0.6\,\text{µm}$) was far less than the measured value. These results illustrated that the underestimation of the pore volume fraction with respect to the smallest pore diameter ($d \leq 0.6\,\text{µm}$) was a key factor with regard to the underestimation of the water content in the dry range of the SWRC. In addition, the underestimation of the pore volume fraction is associated with an oversimplified pore space geometry, which traditional models have generally characterized as a bundle of cylindrical capillaries. The measured and calculated pore curves of the other 46 soil samples behaved in the

same fashion, and those curves are provided in the Supplement (Fig. S1).

## 3 Improved method

### 3.1 Estimating the pore volume fraction

In this study, the soil pore structure was conceptualized within a pore model in which the elementary unit cell is composed of a relatively larger circle-shaped central pore connected to two slit-shaped spaces (see Fig. 2). Relative to the polygonal central pore connected to the slit-shaped spaces as described by Or and Tuller (1999), both the slit width and the slit length are proportional to the diameter of the associated central pore $d$ and are therefore expressed as $\alpha d$ and $\beta d$, respectively.

When estimating the pore volume fraction using the pore model described above, the volume fractions of the central pore and slit-shaped spaces are distinguished. Considering that the sizes of the slit-shaped spaces are smaller than that of the minimum central pore, the slit-shaped spaces are accordingly classified into it. Therefore, the pore volume fractions of the soil samples were simplified into those of the central pores, but the volume fraction of the minimum central pores included that of all slit-shaped spaces. Using the geometric relationship described in Fig. 2 and the traditional assumption that the volume fraction of each unit cell (i.e., the central pore connecting to two slit-shaped spaces) is equal to the corresponding particle mass fraction, the pore volume fractions with respect to different sizes can be readily obtained.

The procedure utilized to calculate the pore volume fractions is shown in Fig. 3. Assuming that the soil pores are composed of numerous unit cells with various sizes, the fraction of the $i$th unit cell is equal to the relative particle mass fraction $\omega_i$. The addition of the volume fraction of the smallest unit cell ($\omega_1$) and the sum of the slit volume fractions of various sizes ($\zeta_2 + \zeta_3 + \ldots + \zeta_i$ TS7) result in the volume fraction of the smallest pore ($\nu_1$). Successively accumulating it with the volume fractions of other central pore (i.e., $\nu_2$, $\nu_3$, $\nu_4$ …) provides the PoSD of a sample. The volume fraction of the slit-shaped spaces, $\zeta_i$, the volume fraction of the smallest pore $\nu_1$ and the volume fractions of the other pores $\nu_i$ were calculated using Eqs. (7), (8) and (9), respectively:

$$\zeta_i = \omega_i \frac{2\alpha\beta d_i^2}{2\alpha\beta d_i^2 + \frac{\pi}{4} d_i^2}, \tag{7}$$

$$\nu_1 = \omega_1 + \sum_2^n \zeta_i, \tag{8}$$

$$\nu_i = \omega_i - \zeta_i, \tag{9}$$

where $\zeta_i$ is the volume fraction of the slit-shaped spaces, $\nu_i$ is the volume fraction of the $i$th pore fraction, and $\alpha$ and $\beta$ are the scaling parameters of the slit width and the slit length, respectively.

### 3.2 Values of $\alpha$ and $\beta$

To obtain the values of $\alpha$ and $\beta$, an expression containing both of these parameters with respect to the specific surface area ($S_{SA}$) was applied here. The $S_{SA}$ of the pore as shown in Fig. 2 can be described using a geometrical relationship as follows:

$$S_{SA} = \frac{\phi}{1000\rho_b} \sum_{i=1}^n \omega_i \frac{4\beta d_i + \pi d_i}{2\alpha\beta d_i^2 + \frac{\pi}{4} d_i^2}; \ i = 1, 2, \ldots n, \tag{10}$$

where $S_{SA}$ is the specific surface area ($\text{m}^2\,\text{g}^{-1}$), $d_i$ is the pore diameter (m), $\rho_b$ is the bulk density ($\text{kg}\,\text{m}^{-3}$) and $\Phi$ is the measured porosity. Therefore, an important requirement for the calculation of the $\alpha$ and $\beta$ values is an estimation of the $S_{SA}$ at sample scale. Here, a power equation was applied as follows (Sepaskhah et al., 2010):

$$S_{SA} = 3.89 d_g^{-0.905}, \tag{11}$$

where $S_{SA}$ is the estimated specific surface area ($\text{m}^2\,\text{g}^{-1}$), and $d_g$ is the geometric mean particle size diameter (mm) obtained using Eq. (12) (Shirazi and Boersma, 1984):

$$d_g = \exp\left(f_c \ln M_c + f_{si} \ln M_{si} + f_{sa} \ln M_{sa}\right), \tag{12}$$

where $f_c$, $f_{si}$ and $f_{sa}$ are the clay, silt and sand fractions (%) of the soil sample, respectively; $M_c$, $M_{si}$ and $M_{sa}$ are the mean diameters of clay, silt and sand that are empirically taken as 0.001, 0.026 and 1.025 mm, respectively.

Consequently, the quantitative relationship between the parameters $\alpha$ and $\beta$ can be obtained using Eq. (10), associated with the additional constraint of Eq. (11), and the values of $\alpha$ and $\beta$ can be theoretically solved if the measured volume fraction of the slit-shaped pore or the measured SWRC is known. However, an analytical solution is difficult to derive due to the high nonlinearity of both equations. Here, a trial and error approach was adopted that was much easier than the analytical method. Conveniently, the UNSODA database provided a great deal of soil information, including the measured SWRCs and diverse soil physical properties.

The routine procedure for handling a soil sample involved the following steps. First, given the initial value of $\alpha$, the value of $\beta$ was calculated using Eqs. (10)–(12), after which the PoSD was predicted using Eqs. (7)–(9). Subsequently, the SWRC was estimated using the method described in Sect. 3.3. Finally, the values of $\alpha$ were changed repeatedly until the newer predicted SWRC was in good agreement with the measured SWRC and the water content corresponding to a suction head of 5000 cmH$_2$O was within 90 % of the measured data (see Fig. S2). The results for the 48 soil samples indicated that the $\beta$ values exhibited a broad variation range for all samples, while the $\alpha$ values showed regular changes with the soil texture. The relationship between the sand contents and the $\alpha$ values for 48 samples is shown in Fig. 4, which clearly demonstrates that the values of $\alpha$ are similar to samples with specific sand contents.

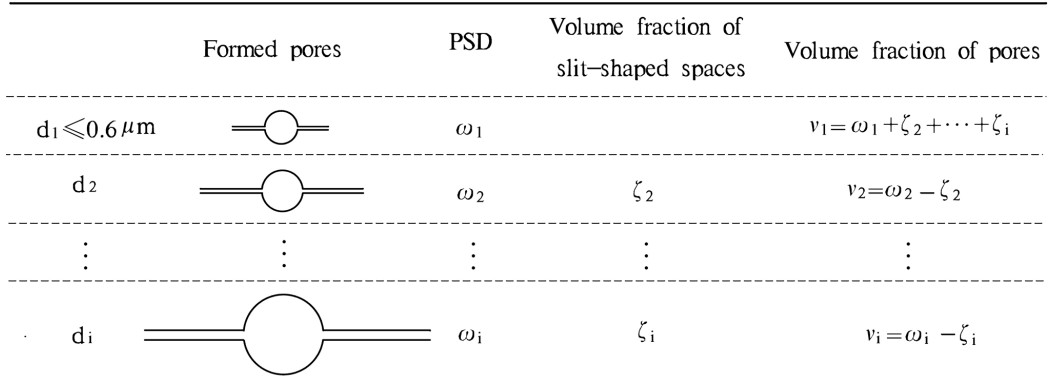

**Figure 3.** Schematic of the procedure used to calculate the pore volume fraction.

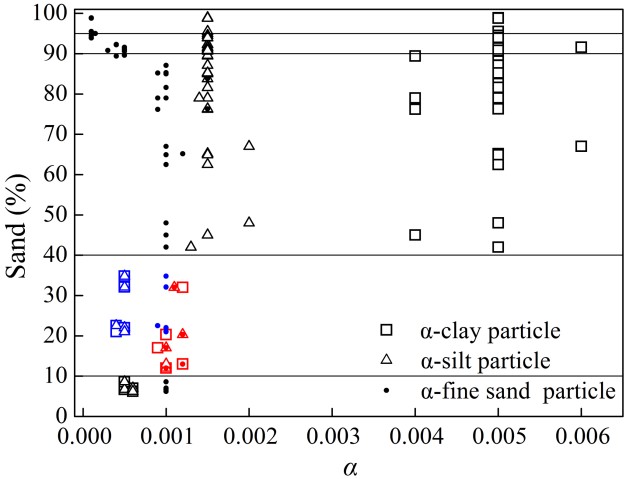

**Figure 4.** The $\alpha$ values for 48 soil samples with different sand content. The $\alpha$ values for clay, silt, and fine sand particles of specific samples are listed in Fig. 4, except those of the coarse sand particle, which are the same value of 0.0004 for all of the samples. For the samples with sand content ranging from 10 % to 40 %, two sets of $\alpha$ value are observed. The $\alpha$ values for the silt content less than and more than 50 % are highlighted in red and blue, respectively, thereby reflecting the dominant functions of the silt or clay particle on the hydraulic properties of typical samples.

Therefore, the approach was simplified by setting $\alpha$ as a constant for similar soil textures. The corresponding detailed descriptions are summarized in Table 3. The values of $\alpha$ were inside the range of $3.34 \times 10^{-05}$ to $2.12 \times 10^{-02}$ TS8, which were estimated by Or and Tuller (1999) using a pore model comprising a polygon-shaped central pore connected to the slit-shaped spaces. According to the sand contents of the samples, Table 3 is a reference for determining the $\alpha$ values that serve as input parameters in predicting the SWRC from the PSD hereafter.

## 3.3 Estimating the SWRC

The values of $\alpha$ and $\beta$ for the various soil samples facilitated the acquisition of the volume fractions of the slit pores using Eq. (7) and the PoSD using Eqs. (8) and (9). The water contents associated with different pore filling stages could be estimated by substituting the PoSD into Eq. (3), and the pore size and the corresponding suction head could be calculated using Eqs. (2) and (5). The SWRC could be ultimately obtained using the calculated suction heads and water contents.

## 4 Model validation

### 4.1 Data sources

Twenty-nine soil samples with a wide range of physical properties were also selected from the UNSODA database to validate the model; the codes of the samples are summarized in Table 4 and their detailed information is presented in Table 5. For the soil samples that were not provided with a saturated water content $\theta_s$, the first data point of the measured SWRC corresponding to the lowest suction head was regarded as $\theta_s$.

To generate a detailed PSD, a modified logistic growth model (Eq. 6) was used to fit the measured PSD data. Here, the detailed PSD was generated at diameter classes of 2, 5, 10, 15, 20, 30, 40, 50, 60, 70, 80, 90, 100, 200, 500, 1000 and 2000 μm. The values of $\alpha$ were chosen from Table 3 according to the sand contents of the soil samples. The values of $\beta$ were obtained by substituting the $S_{SA}$ values predicted using Eq. (11) into Eq. (10). Then, the PoSD was predicted using Eqs. (7)–(9). Finally, the SWRC was estimated using the methods, as described in Sect. 3.3.

The SWRC was also predicted using the traditional method presented in Sect. 2. In the traditional method, the predicted PoSD was equivalent to the PSD (Eq. 1) and was substituted into Eq. (3) to obtain the water content. The corresponding suction heads were predicted using Eqs. (2) and (5).

**Table 3.** The estimated values of $\alpha$ for various soil textures.

| Sand content | Silt content | $\alpha$ | | | |
|---|---|---|---|---|---|
| (%) | (%) | Clay $D \leq 2\,\mu m$ | Silt $2\,\mu m < D \leq 50\,\mu m$ | Fine sand $50\,\mu m < D \leq 500\,\mu m$ | Coarse sand $500\,\mu m < D \leq 2000\,\mu m$ |
| 0–10 | | 0.0005 | 0.0005 | 0.001 | 0.0004 |
| 10–40 | 0–50 | 0.001 | 0.001 | 0.001 | 0.0004 |
| | 50–100 | 0.0005 | 0.0005 | 0.001 | 0.0004 |
| 40–90 | | 0.005 | 0.0015 | 0.001 | 0.0004 |
| 90–95 | | 0.005 | 0.0015 | 0.0005 | 0.0004 |
| 95–100 | | 0.005 | 0.0015 | 0.0001 | 0.0004 |

**Table 4.** Codes of the 29 soil samples selected from UNSODA for the model validation.

| UNSODA codes | Texture class |
|---|---|
| 1360, 4120, 2361, 3282, 1320 | Clay |
| 3190, 1370 | Loam |
| 3160, 3152, 1030, 1090, 4010 | Loamy sand |
| 3155, 3144, 1463, 3132, 4000 | Sand |
| 4620, 4621, 1102, 2341 | Sandy clay loam |
| 3290, 3310 | Sandy loam |
| 4531, 4510 | Silt loam |
| 3031, 3032, 1372, 1362 | Clay loam |

A scaling approach proposed by Meskini-Vishkaee et al. (2014) was used to compare with the proposed method to demonstrate its prediction performance. The detailed calculation procedures were described by Meskini-Vishkaee et al. (2014).

The van Genuchten equation (Eq. 13) was used to fit the predicted SWRC calculated via the three models (van Genuchten, 1980):

$$\frac{\theta - \theta_r}{\theta_s - \theta_r} = \left[ \frac{1}{1 + (a\psi)^n} \right]^m, \tag{13}$$

where $\theta$ is the water content ($cm^3\,cm^{-3}$), $\theta_r$ is the residual water content ($cm^3\,cm^{-3}$), and $a$, $n$, $m$, and $\theta_r$ are fitting parameters. The predicted SWRCs of 29 samples exhibited good fits with an average $r^2$ value of greater than 0.999.

For each set of predictions, the agreement between the predicted and measured water contents was expressed in terms of the root mean square error ($E_{RMS}$), which is given by

$$E_{RMS} = \sqrt{\frac{1}{N} \sum_{i=1}^{N} (\theta_{pi} - \theta_{mi})^2}, \tag{14}$$

where $N$ is the number of measured data points, $\theta_{pi}$ is the predicted water content and $\theta_{mi}$ is the measured water content.

## 4.2 Results

The predicted and measured SWRCs in Fig. 5 show that the improved method exhibited good fits with the measured data in the entire range of the SWRC; moreover, the improved method was clearly better than the traditional method and the scaling approach, especially in the dry range (the other 25 samples are listed in Fig. S3). In this study, the scaling approach, which improved the performance of the original MV–VG model via scaling of the parameter $n$ in the van Genuchten equation, performed better than the traditional method for clay (code: 1360), loamy (code: 3190) and loamy sand (code: 3160). However, it performed worse for coarse-textured soil (e.g., sand (code: 3144)), which may result from the relatively small scaling degree of the parameter $n$ and the poor fit of the fitting equation to the measured PSD data in their study. In general, the improved method here applied well to a wide range of soils, while the scaling approach performed better for fine- and medium-textured soils.

Table 6 shows the $E_{RMS}$ of the improved method, the scaling approach and the traditional method for samples used in model validation. The $E_{RMS}$ values range from 0.017 to 0.054 for the improved method (with an average of 0.028), from 0.026 to 0.060 for the scaling approach (with an average of 0.037) and from 0.040 to 0.106 for the traditional method (with an average of 0.061). In terms of the $E_{RMS}$, the improved method provided the best predictions and the traditional method performed the worst. The results also showed that there is a significant difference between the performance of the improved method and the traditional method ($p = 0.001$). Only for the sand samples does the performance of the improved method and scaling approach exhibit a significant statistical difference ($p = 0.01$).

The accuracy of the predicted SWRC using the improved method depends on the accuracy of the corresponding predicted pore volume fractions. The calculated and measured pore volume fraction curves in Fig. 6 indicate that the predicted curves using the improved method are more similar to the measured data than those predicted using the traditional method, thereby showing that the proposed method performed better. The errors in the predicted pore fractions

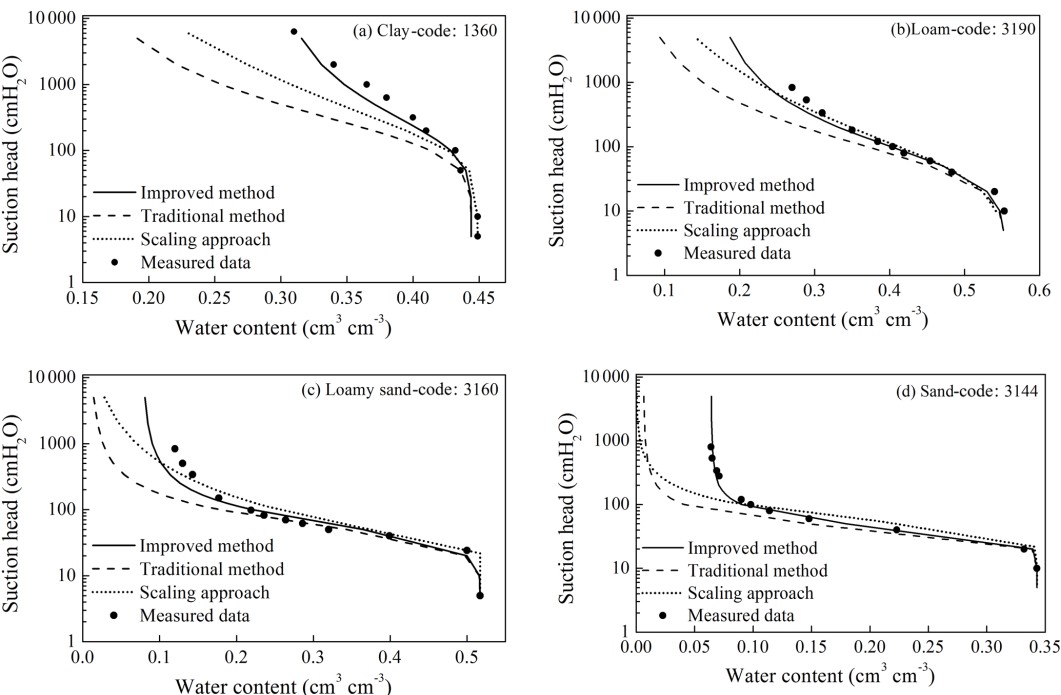

**Figure 5.** Measured and predicted SWRCs for clay (code: 1360), loam (code: 3190), loamy sand (code: 3160) and sand (code: 3144).

**Table 5.** TS9 Basic soil properties of 29 samples for the model validation.

| Soil texture | Number of soil samples | | Clay (%) | Sand (%) | $\rho_b$ (g cm$^{-3}$) |
|---|---|---|---|---|---|
| Clay | 5 | Min | 43.0 | 5.4 | 1.10 |
| | | Max | 57.0 | 32.0 | 1.50 |
| | | Average | 51.0 | 14.4 | 1.31 |
| Loamy | 2 | Min | 16.5 | 47.9 | 1.41 |
| | | Max | 29.2 | 43.6 | 1.45 |
| Loamy sand | 5 | Min | 1.7 | 75.5 | 1.37 |
| | | Max | 7.3 | 85.2 | 1.59 |
| | | Average | 4.9 | 81.0 | 1.46 |
| Sand | 5 | Min | 1.1 | 90.1 | 1.46 |
| | | Max | 4.4 | 97.5 | 1.58 |
| | | Average | 2.3 | 93.4 | 1.53 |
| Sandy loam | 2 | Min | 11.4 | 56.8 | 1.44 |
| | | Max | 12.6 | 65.7 | 1.46 |
| Sandy clay loam | 6 | Min | 9.8 | 28.0 | 1.21 |
| | | Max | 30.7 | 69.7 | 1.53 |
| | | Average | 22.8 | 43.2 | 1.45 |
| | | Min | 33.4 | 20.4 | 1.07 |
| Clay loam | 4 | Max | 37.5 | 34.7 | 1.58 |
| | | Average | 35.1 | 24.8 | 1.27 |

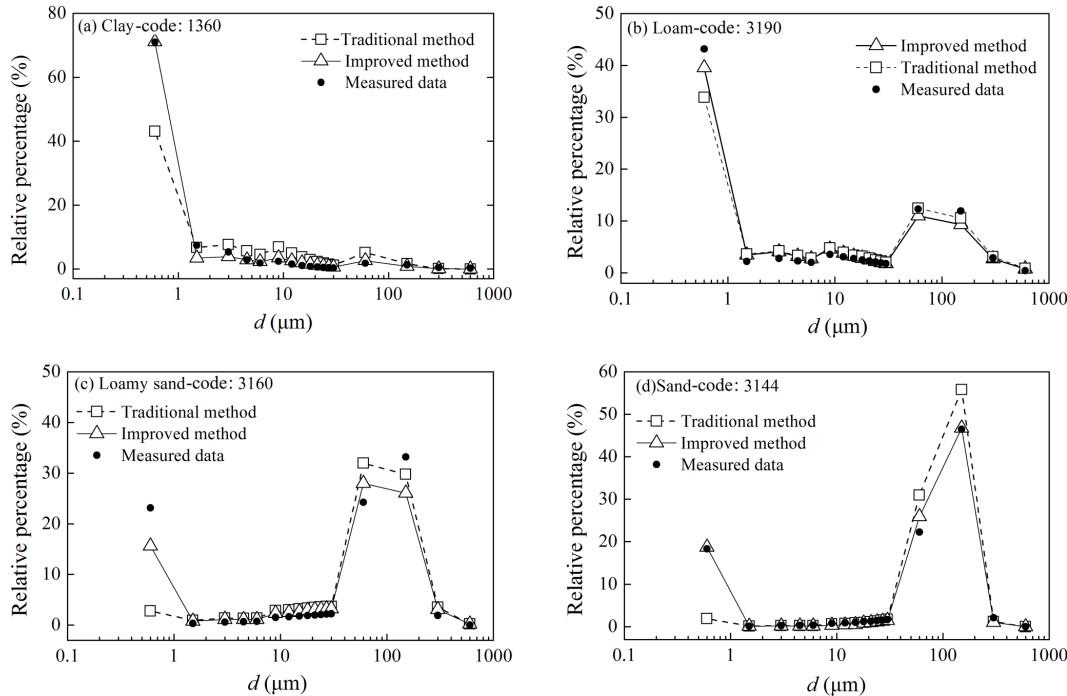

**Figure 6.** The measured and predicted pore volume fraction curves using the improved method and traditional method for clay (code: 1360), loam (code: 3190), loamy sand (code: 3160) and sand (code: 3144).

**Table 6.** The root mean square errors ($E_{RMS}$) of the predicted SWRC using the improved method, the scaling approach and the traditional method.

| Soil texture | Number of soil samples | $E_{RMS}$ | | |
|---|---|---|---|---|
| | | Improved method | Scaling approach | Traditional method |
| Clay | 5 | 0.022 | 0.032 | 0.056 |
| Clay loam | 4 | 0.034 | 0.041 | 0.079 |
| Sandy clay loam | 4 | 0.032 | 0.046 | 0.072 |
| Loam | 2 | 0.054 | 0.060 | 0.106 |
| Loamy sand | 5 | 0.020 | 0.026 | 0.048 |
| Sand | 5 | 0.017 | 0.028 | 0.042 |
| Sandy loam | 2 | 0.046 | 0.049 | 0.068 |
| Silt loam | 2 | 0.024 | 0.031 | 0.040 |

using the traditional method mainly occur at the minimum pore size ($d \le 0.6 \, \mu m$), which proves that the errors of the predicted SWRC using the traditional method originate from the neglect of small pores, such as slit-shaped spaces in a natural sample. The proposed method used the pore model containing slit-shaped spaces to represent the pore space geometry and consequently improved the prediction of the SWRC. However, the uncertainties are unavoidable when choosing the parameters $\alpha$ for unknown media, which is the main factor affecting the accuracy of the predicted SWRCs.

### 4.3 Discussion

#### 4.3.1 The suction head calculation in the slit-shaped spaces

When capillary water coexists with adsorptive water in the narrow pores, the capillary and surface forces, including ionic–electrostatic, molecular, structural, and adsorption forces, contribute to the potential energy of water in the slit-shaped pores (Tuller et al., 1999; Iwamatsu and Horii, 1996). When considering only the capillary forces, the drainage potential in slit-shaped pores is given as Eq. (15) (Derjaguin and Churaev, 1992):

$$\mu = \frac{-2\sigma}{\rho \alpha d}, \tag{15}$$

where $\mu$ is the critical potential ($J \, kg^{-1}$).

However, the applicability of this formula is limited by the width of the slit. Tuller and Or (2001) defined a critical slit spacing ($\alpha d^*$) by Eq. (16) that classifies the slit sizes responding to capillary drainage and adsorption-dominated drainage. In the case of slit-shaped spaces greater than $\alpha d^*$, the capillary-based slit drainage is applied.

$$\alpha d^* = \sqrt{-\frac{9 A_{svl}}{4\pi \sigma}}, \tag{16}$$

where $A_{svl}$ is the Hamaker constant for solid–vapor interactions through the intervening liquid, usually set as $-6.0 \times$

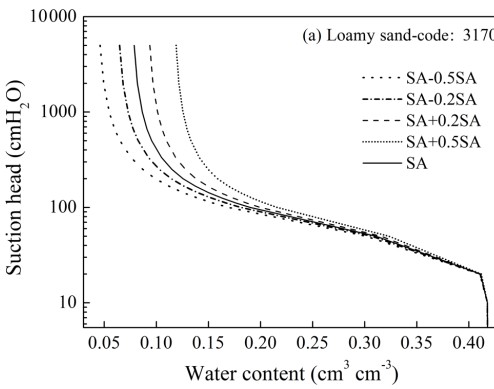
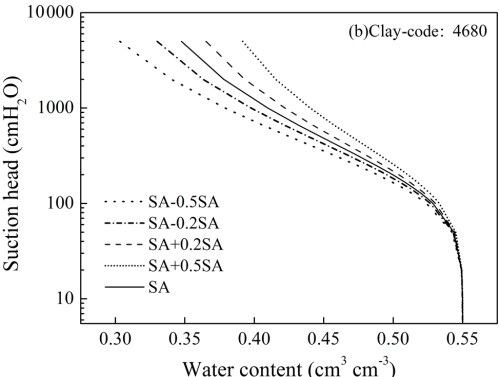

**Figure 7.** The effects of a change in the estimated $S_{SA}$ on the SWRC for loamy sand (code: 3170) and clay (code: 4680). The $S_{SA}$ denotes the accurate value of the specific surface area.

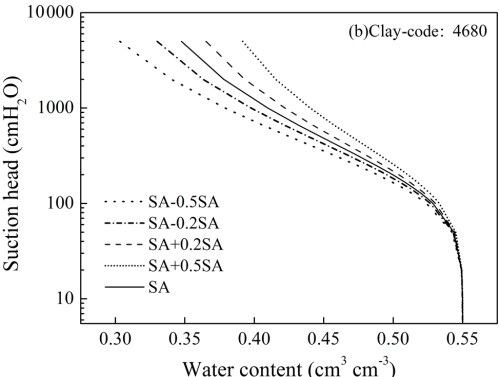

**Figure 8.** The calculated slit width $\alpha d$, slit length $\beta d$ and $S_{SAi}$ for loamy sand (code: 3170).

$10^{-20}$ J [TS10] (Tuller and Or, 2001). The value of $\alpha d^*$ is 0.591 nm, which means that Eq. (15) could be applied to calculate the drainage potential in the slit-shaped spaces in this study.

In our study, the critical drainage suction head for the minimum central pore calculated using Eq. (5) is 5000 cmH₂O, while that of the widest slit-shaped spaces calculated using Eq. (15) is 6202 cmH₂O (the potential is converted to the suction head). This result illustrates that all slit-shaped spaces are still filled with water when the suction head is up to the critical drainage suction head for the minimum central pores. On the other hand, the largest slit width calculated from the parameters in Table 3 is 0.24 μm, which is smaller than the minimum pore diameter of 0.6 μm. According to the above analysis, it is reasonable that the volume fractions of the minimum pores include the volume fractions of the minimum central pore and all slit-shaped spaces.

### 4.3.2   The effects of the estimated $S_{SA}$

The $S_{SA}$ values estimated using Eq. (11) could affect the accuracy of the predicted SWRCs. Figure 7 shows that an overestimation of the $S_{SA}$ prompts the dry range of the SWRC to move in the direction of larger water content, and vice versa. When the estimated $S_{SA}$ value was altered by 10 % and −10 % of its accurate value for loamy sand (code: 3170), the water contents with respect to the highest suction head were higher and lower, respectively, by approximately $0.007\,\text{cm}^3\,\text{cm}^{-3}$ than those of the original SWRC. For clay (code: 4680), the water contents were higher and lower by approximately $0.009\,\text{cm}^3\,\text{cm}^{-3}$ at the same 10 % and −10 % alterations, respectively. Consequently, for coarse-textured soil, the water content and prediction error of the SWRC changed relatively little for the same degree of change in the $S_{SA}$. Figure 7 also showed that a relatively small error appeared between the calculated and measured SWRCs when the error of the estimated $S_{SA}$ was within 20 %.

Previous work has shown that the $S_{SA}$ of soil is closely dependent upon the soil texture and could be estimated from the soil properties and PSD (Sepaskhah and Tafteh, 2013; Resurreccion et al., 2015). The method used to estimate the $S_{SA}$ in Sect. 3.2 was presented by Sepaskhah et al. (2010), who estimated the $S_{SA}$ based on the geometric mean particle size diameter as shown in Eq. (12) with an $r^2$ value of 0.88. Moreover, the appropriateness of this equation was validated using 64 soil samples by Fooladmand (2011). Sepaskhah et al. (2010) pointed out that the estimation deviations increased distinctly for measured $S_{SA}$ greater than $200\,\text{m}^2\,\text{g}^{-1}$. In the proposed method, the estimated $S_{SA}$ is mainly used to gain the parameter $\alpha$ and $\beta$ and to estimate the volume fractions of the slit-shaped spaces; thus, the estimation accuracy of $S_{SA}$ influences the dry range of the SWRC (Fig. 7) and equivalently influences the degree of improvement in the predicted SWRC. Therefore, more effort should be made toward developing a more accurate transformation from the

soil physical properties to $S_{SA}$ to further improve the prediction of the SWRC.

### 4.3.3 The slit-shaped spaces and the $S_{SA}$ at the sample scale

Since the central pore diameter $d$ is proportional to the corresponding particle diameter $D$, the slit width $\alpha d$, the slit length $\beta d$ and the specific surface area $S_{SAi}$ of each unit cell are associated with the particle size. The calculated values of $\alpha d$, $\beta d$ and $S_{SAi}$ of clay, silt, fine sand and coarse sand particles for the loamy sand (code: 3170) are listed in Fig. 8. The results confirm that the pores formed by bigger soil particles are large with a correspondingly large slit width $\alpha d$; this is similar to the results of Or and Tuller (1999), and the values are on the same order of magnitude. It is common knowledge that larger soil particles tend to have large surface areas, and therefore, the slit length formed by the contact of soil particle edges should be relatively long, leading to the positive relationship between the slit length $\beta d$ and the particle diameter as shown in Fig. 8. This result is different from that in Or and Tuller (1999), where the slit length $\beta d$ was inversely proportional to the particle diameter. In addition, the $S_{SAi}$ of the $i$th particle fractions decreased with an increase in the particle diameter, which is consistent with the findings of Or and Tuller (1999) and is in accordance with the general understanding of the $S_{SA}$.

## 5 Conclusions

The traditional models employed to translate the PSD into the SWRC underestimate the water content in the dry range of the SWRC. The errors originate from a setting that the cumulative PoSD is equal to the corresponding PSD, which resulted in an underestimate of the pore volume fraction of the minimum pore diameter range and consequently the water content in the dry range of the SWRC. If slit-shaped pore spaces are taken into consideration when estimating the PoSD with a pore model comprising a circle-shaped central pore connected to slit-shaped spaces, the pore volume fraction of the minimum pore diameter range will be accordingly increased; therefore, the SWRC can be more accurately predicted. The estimation of the $\alpha$ and $\beta$ values is a key step to predicting the SWRC in the proposed method. The $\alpha$ values were obtained using 48 measured soil samples, and those values served as input parameters for predicting the SWRC; then, the $\beta$ values were readily calculated using a constraint on the estimated $S_{SA}$. The validation results illustrate that the SWRCs predicted using the proposed method provided the best predictions of the SWRCs, closely followed by the scaling approach, and the traditional method performed worst.

*Data availability.* The unsaturated soil hydraulic properties and basic soil properties of samples are available from the UNSODA database (https://www.ars.usda.gov/research/datasets/ TS11 ).

**The Supplement related to this article is available online at https://doi.org/10.5194/hess-22-1-2018-supplement.**

*Author contributions.* CC TS12 developed the method and wrote the paper. DC performed the analysis and contributed ideas and comments on the method and writing.

*Competing interests.* The authors declare that they have no conflict of interest.

*Acknowledgements.* This research was partially supported by the Special Fund for Basic Scientific Research of Central Colleges (310829162015) and by the National Natural Science Foundation of China (41472220). The authors thank Kang Qian for providing the UNSODA unsaturated soil hydraulic property database.

Edited by: Roberto Greco
Reviewed by: Fatemeh Meskini-Vishkaee and one anonymous referee

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

**Remarks from the typesetter**