# Peer review of "Predicting the soil water characteristic curve from the particle size distribution based on a pore space geometry containing slit-shaped spaces"

_Hydrology and Earth System Sciences, 2017_

## Referee Comment (RC1) · F. Meskini-Vishkaee (Referee) · 17 Jan 2018

The submitted paper introduces an improved model to estimate soil water retention curve (SWRC) from soil particle size distribution (PSD) data that based on a pore space geometry containing slit-shaped spaces. However, predictions of improved method were more accurate than those of Arya model, but this superiority may be caused by some assumptions and simplifications.

1. Since The relationship between the PSD and the pore size distribution (PoSD) is a

fundamental element when predicting the SWRC from the PSD, first adjective of this study was to compare estimated PoSD using traditional Method to measured PoSD.

a. This step includes i, estimated PoSD from PSD and ii, estimated PoSD from SWRC. The authors have to change subtitle "2) measuring the PoSD" in page 3, line 33 by "2) estimating the PoSD from SWRC".

b. To estimate PoSD from PSD, called the traditional method as Arya model, here a proportionate relationship between pore size and associated particle diameter was used to calculate the equivalent pore diameter (Eqn. 2) because it was easy to use. This simplification may be a part of the estimation error of Arya and Paris (1891) model.

c. It is noted that estimation method of PoSD from SWRC is nearly similar to the estimation method of PoSD from PSD proposed by Mohammadi and Vanclooster (2010). Although, since SWRC is influenced both soil texture and structure, if soil organic carbon or clay content would be high, differences between estimated PoSD from SWRC and PSD become more. It must be mentioned that the prediction error of estimated SWRC from PSD is at dry range of SWRC (at high suction heads) that influences by soil texture (especially clay particles). Mohammadi and Meskini-Vishkaee (2012) attribute the methods error to the roughness of soil particles, high surface energy content of clay particles and, to the simplified pore geometric concepts that does not effectively reflect the pore geometry. It is better that the authors compare estimated PoSD from measured SWRC to estimated PoSD from PSD using similar method (use Mohammadi and vanclooster method as traditional method). Therefore, I think that these calculations have to add to this part of manuscript.

2. Tuller et al. (1999) and Or and Tuller (1999) proposed including the water films coating the pore walls and water in angular spaces of pores, in calculations of soil water content. Despite great scientific interest, the proposed approach for the derivation of SMC by Or and Tuller (1999) motivated by bundle of cylindrical tubes limitations, usually fails to describe experimental data in the intermediate soil water content range

because of the low flexibility of the gamma distribution function used to characterize the PoSD (Lebeau and Konrad, 2010). In addition, the model is mathematically complex and furthermore needs specific surface area parameter which measurements and estimations are often quite variable (Carter et al., 1986).

a. The authors use pore geometry containing slit-shaped spaces proposed by and Or and Tuller (1999), But they assumed that circle-shaped central pore connected to two slit-shaped spaces. Moreover, the estimated PoSD data were fitted using a modified logistic growth model (Eqn. 5).

b. Specific surface area (SSA) is a requirement parameter to obtain the values of a and B. The authors used a power equation with two fitting parameters (Eqn. 10) to estimate SSA proposed by Sepaskhah et al. (2010). Sepaskhah et al. (2010) used twenty soil samples from a depth of 0–30 cm were collected from different locations in Fars province, in the south of Iran to calibrate the power equation. In addition, a different set of data was used to validate the calibrated model. Their results indicated that in the range of around 20 up to 200 m2 g-1 the values of measured SSA were in quite a good agreement, while for SSA greater than 200 m2 g-1, the deviations increase distinctly. Moreover, Tuller and Or (2005) stated that the psychrometric approach for SSA determination should provide reliable values for natural soils with hydratable surface areas below 200 m2/g. They recommend using SWRC values for -10 MPa and lower (drier) with an effective Hamaker constant of -6 ×10-20 J to predict SSA values. So, there are some ambiguities here,

i. As respects higher SSA is related to finer texture soils that usually have underestimation problem of estimated SWRC from PSD, Indeed, I think use power model to estimate SSA cannot be useful to improve estimated SWRC in fine-textured soils.

Page 9, line 4: the authors declared that "for the coarse-textured soil, the water content and prediction error of the SWRC changed relatively little for the same degree of change of the SSA". This is completely expected because not only there is not serious

problem to estimate SWRC from PSD in coarse-textured soils, but also the value of estimated SSA using power equation is below 200 m2 g-1 for coarse-textured soils.

ii. Is there any SSA measurement? Were the fitting parameters of power model controlled?

3. At the first step, the estimated PoSDs of 48 soil samples using SWRC were compared with the PoSDs calculated using PSD to identify the origins of the errors and their effects on the accuracy of the SWC and to calibrate the proposed model. Subsequently, 22 soil samples were also selected from UNSODA database to validate the model.

a. Please provide a Table involved some properties of selected samples for both calibration and validation data sets (e.g. max, min and average of clay content, organic matter, bulk density and . . . for each soil textural class).

b. About validation data set, Textural distribution of the 22 soil samples is shown in both Figure 5 and Table 3. This duplication is not necessary.

c. As regards the most prediction error of traditional models is often related to soils with good structure or high clay content. Therefore, the authors have to use more fine-textured soils to validate their proposed model. In validation data set, only 4 soil samples had clay texture and more than 60 % of soil samples are coarse-textured soils! Please add more soil samples with higher clay content and organic matter to the validation data set.

4. In page 8, line 19-21: the authors stated that "These improvements are mainly attributed to the pore model containing slit-shaped spaces, demonstrating that this pore model is better for predicting the SWC from the PSD than the concept of a bundle of cylindrical tubes". This simplification (concept of a bundle of cylindrical tubes) is introduces as major source of error in the SWRC predictor models using PSD. After that, some studies have attempted to improve the water content calculation approach by at-

tributing model errors to both a simplified pore geometry and an incomplete desorption of residual water in the soil pore within the high matric suction head range. Therefore, I think the authors have to compare proposed model to other models except Arya and Paris (1981), such as Mohammadi and Meskini-Vishkaee (2012) or Meskini-Vishkaee et al. (2014) or other models. The comparison between the performance of these models and parameter needs can be more helpful.

Please expand discussion part and state the result of proposed model for both data sets (calibration and validation) in more detail.

References

Arya, L. M., and Paris, J. F.: A physicoempirical model to predict the soil moisture characteristic from particle-size distribution and bulk density, Soil Science Society of America Journal, 45, 1023-1030, 1981.

Lebeau, M., Konrad, J.-M. A new capillary and thin film flow model for predicting the hydraulic conductivity of unsaturated porous media. Water Resour. Res. 46, W12554. 2010.

Meskini-vishkaee, F., Mohammadi, M. H., and Vanclooster, M.: Predicting the soil moisture retention curve, from soil particle size distribution and bulk density data using a packing density scaling factor, Hydrology & Earth System Sciences, 18, 4053-4063, 2014.

Mohammadi, M. H., and Meskini-Vishkaee, F.: Predicting the film and lens water volume between soil particles using particle size distribution data, Journal of Hydrology, 475, 403-414, 2012.

Mohammadi, M. H., and Vanclooster, M.: Predicting the soil moisture characteristic curve from particle size distribution with a simple conceptual model, Vadose Zone Journal, 10(2), 594-602, 2011.

Or, D., and Tuller, M.: Liquid retention and interfacial area in variably saturated porous

media: Upscaling from single-pore to samplescale model, Water Resources Research, 35, 3591-3605, 1999.

Sepaskhah, A. R., Tabarzad, A., and Fooladmand, H. R.: Physical and empirical models for estimation of specific surface area of soils, Archives of Agronomy & Soil Science, 56, 325-335, 2010.

Tuller, M., and Or, D.: Water films and scaling of soil characteristic curves at low water contents, Water Resources Research, 41, 319-335, 2005.

Tuller, M., Or, D., and Dudley, L. M.: Adsorption and capillary condensation in porous media: Liquid retention and interfacial configurations in angular pores, Water resources Research, 35, 1949–1964, 1999.

---

## Author Comment (AC1) · 15 Feb 2018

Dear Meskini-Vishkaee,

We thank you for your time and the constructive comments. The following are our responses to your comments. We expected more suggestions from you to improve our manuscript.

Regards

[Figure]

Cheng Dong-hui Hydrology professor

1. Since the relationship between the PSD and the pore size distribution (PoSD) is a fundamental element when predicting the SWRC from the PSD, first adjective of this study was to compare the estimated PoSD using traditional Method with the measured PoSD. The following comments and responding responses is about PoSD and PSD.

Comment a: This step includes i, estimated PoSD from PSD and ii, estimated PoSD from SWCC. The authors have to change subtitle "2) measuring the PoSD" in page 3, line 33 by "2) estimating the PoSD from SWCC".

Response: We will revise the subtitle mentioned above in revised manuscript.

Comment b: To estimate PoSD from PSD, called the traditional method as Arya model, here a proportionate relationship between pore size and associated particle diameter was used to calculate the equivalent pore diameter (Eqn. 2) because it was easy to use. This simplification may be a part of the estimation error of Arya and Paris (1891) model.

Response: The proportionate relationship between the pore size and the associated particle size was used to calculate the equivalent pore diameter and the suction head in our manuscript (Jensen et al., 2015). In order to evaluate the power of this calculation of suction head, we calculated suction head using proportionate relationship between the pore size and the associated particle size, and the water content calculation followed the way of Arya and Paris (1981) model (Arya and Paris, 1981). Based on the calculated suction and water content, the corresponding SWCC was predicted. Comparing with SWCC predicted using method in Mohammadi and Vanclooster (2011) (Figure 1) (Mohammadi and Vanclooster, 2011), we conclude that the method of calculated suction head using proportionate relationship between the pore size and the associated particle size not only could get a good predicted performance as the method in Mohammadi and Vanclooster (2011), but also was easy to use. In addition, this calculation method of suction head was difference from that in Arya and Paris (1891)
model.

Figure 1: Predicted SWCCs using two methods and measured SWCC

Comment c: It is noted that estimation method of PoSD from SWCC is nearly similar to the estimation method of PoSD from PSD proposed by Mohammadi and Vanclooster (2010). Although, since SWCC is influenced both soil texture and structure, if soil organic carbon or clay content would be high, differences between estimated PoSD from SWCC and PSD become more. It must be mentioned that the prediction error of estimated SWCC from PSD is at dry range of SWCC (at high suction heads) that influences by soil texture (especially clay particles). Mohammadi and Meskini-Vishkaee (2012) attribute the methods error to the roughness of soil particles, high surface energy content of clay particles and, to the simplified pore geometric concepts that does not effectively reflect the pore geometry. It is better that the authors compare estimated PoSD from measured SWCC to estimated PoSD from PSD using similar method (use Mohammadi and vanclooster method as traditional method). Therefore, I think that these calculations have to add to this part of manuscript.

Response: We agree with the reviewer's viewpoint that the error of predicted SWCC from PSD is in dry range of SWCC (at high suction heads) that is influenced by soil texture (especially clay particles). The detailed description about this point of view has been presented in Section 2 "Basic descriptions" of our manuscript.

2. Tuller et al. (1999) and Or and Tuller (1999) proposed including the water films coating the pore walls and water in angular spaces of pores, in calculations of soil water content. Despite great scientific interest, the proposed approach for the derivation of SMC by Or and Tuller (1999) motivated by bundle of cylindrical tubes limitations, usually fails to describe experimental data in the intermediate soil water content range because of the low flexibility of the gamma distribution function used to characterize the PoSD (Lebeau and Konrad, 2010). In addition, the model is mathematically complex and furthermore needs specific surface area parameter which measurements and
estimations are often quite variable (Carter et al., 1986). The following two comments and responses are related to the pore geometry model and its parameters.

Comment a: The authors use pore geometry containing slit-shaped spaces proposed by Or and Tuller (1999), But they assumed that circle-shaped central pore connected to two slit-shaped spaces. Moreover, the estimated PoSD data were fitted using a modified logistic growth model (Eqn. 5).

Response: We use for reference the pore geometry containing slit-shaped spaces proposed by Or and Tuller (1999) (Or and Tuller, 1999), and the detailed information just like the description in comment a.

Comment b: Specific surface area (SSA) is a requirement parameter to obtain the values of $\alpha$ and $\beta$. The authors used a power equation with two fitting parameters (Eqn. 10) to estimate SSA proposed by Sepaskhah et al. (2010). Sepaskhah et al. (2010) used twenty soil samples from a depth of 0–30 cm were collected from different locations in Fars province, in the south of Iran to calibrate the power equation. In addition, a different set of data was used to validate the calibrated model. Their results indicated that in the range of around 20 up to 200 m2 g-1 the values of measured SSA were in quite a good agreement, while for SSA greater than 200 m2 g-1, the deviations increase distinctly. Moreover, Tuller and Or (2005) stated that the psychrometric approach for SSA determination should provide reliable values for natural soils with hydratable surface areas below 200 m2 g-1. They recommend using SWCC values for -10 MPa and lower (drier) with an effective Hamaker constant of -6×10-20 J to predict SSA values. So, there are some ambiguities here, i. As respects higher SSA is related to finer texture soils that usually have underestimation problem of estimated SWCC from PSD, Indeed, I think use power model to estimate SSA cannot be useful to improve estimated SWCC in fine-textured soils. Page 9, line 4: the authors declared that "for the coarse-textured soil, the water content and prediction error of the SWCC changed relatively little for the same degree of change of the SSA". This is completely expected because not only there is not serious problem to estimate SWCC from PSD
in coarse-textured soils, but also the value of estimated SSA using power equation is below 200 m2 g-1 for coarse-textured soils. ii. Is there any SSA measurement? Were the fitting parameters of power model controlled?

Response: The errors generated inevitably when calculating the specific surface area from a regressive power equation (Eq.(5)) (Sepaskhah et al., 2010); however, direct measurements of the specific surface area is difficult and the errors would also exist. Indeed, the specific surface area calculated using a power equation have relative large deviations for SSA greater than 200 m2 g-1. In our manuscript, initially the estimated SSA is combined with the measured SWCC to gain the value of parameter $\alpha$ and $\beta$, and then these parameters were used as input parameters in the predicting SWCC process hereafter. For the predicted SWCCs of fine-textured soils which calculated from the parameter $\alpha$ and $\beta$ and estimated SSA, the errors from estimated SSA, to some extend, could been offset by the parameter $\alpha$ and $\beta$. Certainly, more effort should be directed to a more accurate method of SSA estimation. For fine-textured soils, the SWCC predicted using proposed method fit the measured data well and is better than that predicted using traditional method. Although the error generate when using power model to estimate SSA, the proposed method is useful to improve predicted SWCC in fine-textured soils. For coarse-textured soils, because the SSA of this soil is relative small, the water content and prediction error of the SWCC changed relatively little under the same proportional change of the SSA. We regret that we did not conduct SSA measurement. The power equation employed to predict SSA in our manuscript is an empirical equation and the parameters referred to Sepaskhah et al. (2010).

3. At the first step, the estimated PoSDs of 48 soil samples using SWRC were compared with the PoSDs calculated using PSD to identify the origins of the errors and their effects on the accuracy of the SWC and to calibrate the proposed model. Subsequently, 22 soil samples were also selected from UNSODA database to validate the model. The following three comments and responses are related to the data sets of the soil samples

Comment a: Please provide a Table involved some properties of selected samples for both calibration and validation data sets (e.g. max, min and average of clay content, organic matter, bulk density and......for each soil textural class).

Response: The detailed information for both validation and calibration data sets are presented in Figure 2 and Figure 3, respectively.

Figure 2: Basic soil properties of 22 samples for the model validation.

Figure 3: Basic soil properties of 48 samples for the model calibration.

Comment b: About validation data set, Textural distribution of the 22 soil samples is shown in both Figure 5 and Table 3. This duplication is not necessary.

Response: Figure 5 exhibits more detailed content of clay, silt and sand particle for soil samples in model validation. The Figure 5 may be deleted in order to avoid duplication.

Comment c: As regards the most prediction error of traditional models is often related to soils with good structure or high clay content. Therefore, the authors have to use more fine-textured soils to validate their proposed model. In validation data set, only 4 soil samples had clay texture and more than 60 % of soil samples are coarse-textured soils. Please add more soil samples with higher clay content and organic matter to the validation data set.

Response: The absolute errors of predicted water content to measured water content using traditional models for fine-textured is higher than that for coarse-textured soil. However the relative errors of both fine-textured and coarse-textured soils cannot be ignored. We will add some new soil samples in order to fully validate the predicted model.

4. The following comment is related to the calibration and validation of proposed model.

Comment: In page 8, line 19-21: the authors stated that "These improvements are mainly attributed to the pore model containing slit-shaped spaces, demonstrating that
this pore model is better for predicting the SWC from the PSD than the concept of a bundle of cylindrical tubes". This simplification (concept of a bundle of cylindrical tubes) is introduces as major source of error in the SWCC predictor models using PSD. After that, some studies have attempted to improve the water content calculation approach by attributing model errors to both a simplified pore geometry and an incomplete desorption of residual water in the soil pore within the high matric suction head range. Therefore, I think the authors have to compare proposed model to other models except Arya and Paris (1981), such as Mohammadi and Meskini-Vishkaee (2012) or Meskini-Vishkaee et al. (2014) or other models. The comparison between the performance of these models and parameter needs can be more helpful. Please expand discussion part and state the result of proposed model for both data sets (calibration and validation) in more detailãĂĆ

Response: The traditional and the proposed model used for the SWCC prediction are described in our manuscript in detail. Therefore we did not add the calculation method and the prediction results of other improved model to avoid the excessive content. But we agree that it will improve the manuscript if the performances comparison between the proposed model and other improved models were added. Thus, we will compare our model with the improved model in Meskini-Vishkaee et al. (2014) (Meskini-Vishkaee et al., 2014) and add relative predicted results and discussions.

References

Arya, L. M., and Paris, J. F.: A physicoempirical model to predict the soil moisture characteristic from particle-size distribution and bulk density, Soil Science Society of America Journal, 45, 1023-1030, doi:10.2136/sssaj1981.03615995004500060004x, 1981. Jensen, D. K., Tuller, M., Jonge, L. W. D., Arthur, E., and Moldrup, P.: A new Two-Stage Approach to predicting the soil water characteristic from saturation to oven-dryness, Journal of Hydrology, 521, 498-507, doi: 10.1016/j.jhydrol.2014.12.018, 2015. Meskini-vishkaee, F., Mohammadi, M. H., and Vanclooster, M.: Predicting the soil moisture retention curve, from soil particle size distribution and bulk density data using a packing density scaling factor, Hydrology & Earth System Sciences, 18, 4053-4063, doi: 10.5194/hess-18-4053-2014, 2014. Mohammadi, M. H., and Vanclooster, M.: Predicting the soil moisture characteristic curve from particle size distribution with a simple conceptual model, Vadose Zone Journal, 10(2), 594-602, doi:10.2136/vzj2010.0080, 2011. Or, D., and Tuller, M.: Liquid retention and interfacial area in variably saturated porous media: Upscaling from single-pore to sample-scale model, Water Resources Research, 35, 3591-3605, doi:10.1029/1999WR900262, 1999. Sepaskhah, A. R., Tabarzad, A., and Fooladmand, H. R.: Physical and empirical models for estimation of specific surface area of soils, Archives of Agronomy & Soil Science, 56, 325-335, doi: 10.1080/03650340903099676, 2010.

[Figure]

**Fig. 1.**

**Figure 2: Basic soil properties of 22 samples for the model validation.**

| Soil texture | Number of soil | | Clay (%) | Sand (%) | $\rho_b$ (g m$^{-3}$) |
|---|---|---|---|---|---|
| Clay | 4 | Min | 43 | 5.4 | 1.1 |
| | | Max | 57 | 32 | 1.5 |
| | | Average | 51 | 16.5 | 1.3 |
| Loamy | 2 | Min | 16.5 | 47.9 | 1.41 |
| | | Max | 29.2 | 43.6 | 1.45 |
| Loamy sand | 5 | Min | 1.7 | 75.5 | 1.37 |
| | | Max | 7.3 | 85.2 | 1.59 |
| | | Average | 4.9 | 81 | 1.46 |
| Sand | 5 | Min | 1.1 | 90.1 | 1.46 |
| | | Max | 4.4 | 97.5 | 1.58 |
| | | Average | 2.3 | 93.4 | 1.53 |
| Sandy loam | 2 | Min | 11.4 | 56.8 | 1.44 |
| | | Max | 12.6 | 65.7 | 1.46 |
| Sandy clay loam | 4 | Min | 9.8 | 28 | 1.21 |
| | | Max | 26 | 41 | 1.53 |
| | | Average | 21 | 35.5 | 1.36 |

**Fig. 2.**

**Figure 3: Basic soil properties of 48 samples for the model calibration.**

| Soil texture | Number of soil | | Clay (%) | Sand (%) | $\rho_b$ (g m$^{-3}$) |
|---|---|---|---|---|---|
| Clay | 9 | Min | 41.5 | 6.1 | 1.08 |
| | | Max | 58.2 | 36 | 1.64 |
| | | Average | 50.2 | 14.1 | 1.29 |
| Loam | 4 | Min | 14 | 42 | 1.36 |
| | | Max | 23 | 67 | 1.63 |
| | | Average | 17.3 | 50.5 | 1.46 |
| Loamy sand | 10 | Min | 3 | 76.2 | 1.32 |
| | | Max | 10.4 | 89.4 | 1.6 |
| | | Average | 6.1 | 83.2 | 1.46 |
| Sand | 16 | Min | 0.7 | 89.6 | 1.41 |
| | | Max | 4.6 | 98.9 | 1.7 |
| | | Average | 2.5 | 93.4 | 1.55 |
| Sandy clay loam | 1 | | 2.7 | 62.5 | 1.7 |
| Sandy loam | 3 | Min | 10.5 | 64.9 | 1.27 |
| | | Max | 19.4 | 76.3 | 1.7 |
| | | Average | 15 | 68.8 | 1.50 |
| Silt loam | 5 | Min | 10.5 | 21 | 1.49 |
| | | Max | 15.7 | 34.8 | 1.56 |
| | | Average | 12.6 | 26.5 | 1.52 |

Fig. 3.

---

## Referee Comment (RC2) · Anonymous Referee #2 · 14 Apr 2018

General comment The paper deals with the estimation of soil water retention curve from soil particle size distribution, by means of a geometric model of pores allowing to link solid particle fractions to pore volume fractions. Such a topic can be of interest for some readers of HESS. The paper is concise and clearly organized. The English language is understandable, but would benefit of the help of a native speaker. Although the proposed pore space model is in the end as light modification of former models, the results obtained for a wide variety of soils seem good. Some minor issues should be addressed (see following detailed comments). Once done, I think the manuscript

could be published.

Detailed comments Equation (4) should be rewritten in a more general way, regardless of the units adopted for the water potential. In this regard, it seems that this equation is used to link pore dimension to water potential, even in the silt-shaped space between pores. This aspect should be better clarified, as the dimension of the silts are proportional to the pore diameter, so it is not clear what is the diameter introduced in equation (4) to obtain the corresponding potential. There is also another point, regarding silt-shaped spaces, that in my opinion deserves to be discussed in the paper. To my best understanding, silt-shaped spaces are introduced to consider the water which is bonded to the particles in such a way that the model of the bundle of cylinders fails in describing it. In fact, with such silts dimensions as small as 1 ïA■m are reached. In such a range of dimensions, capillarity is not anymore the mechanism which bonds water to the soil particles, and other kinds of interactions contribute to the potential energy of water (actually, already for quite larger pore dimensions). So, if equation (4) is still used, this turns out to be an effective, but not physically based, way to obtain water potential. Pag. 5, line 13. The water potential values should be negative. Pag. 9, lines 17-19. This statement sounds surprising, if I understand it correctly. The smaller the particles, the larger I expect soil (specific) surface area, as for instance for clay particles. In this respect, the authors should try (where possible), or at least mention the possibility of using measured surface areas rather than estimating it by means of an empirical formula, and discuss how their results could be (positively or negatively) affected. Pag. 10, line 21. The reference should read "van Genuchten, M. T." instead of "Genuchten, M. T. V.", and the same holds for where such a reference is recalled in the text.

---

## Author Comment (AC2) · 3 May 2018

Dear Reviewer,

We appreciate very much for your positive and constructive comments and suggestions on our manuscript entitled "Predicting the soil water characteristic curve from the particle size distribution based on a pore space geometry containing slit-shaped spaces". Now we made a detailed response for your comments as following:

Comment 1: Detailed comments Equation (4) should be rewritten in a more general

way, regardless of the units adopted for the water potential. In this regard, it seems that this equation is used to link pore dimension to water potential, even in the silt-shaped space between pores. This aspect should be better clarified, as the dimension of the silts are proportional to the pore diameter, so it is not clear what is the diameter introduced in equation (4) to obtain the corresponding potential.

Reply to comment 1: Equation (4) in our main manuscript was gained by substituting known parameters into Laplace's equation (Eq. (1))(Haverkamp and Parlange 1986), in which $\sigma$=7.275×10-2 kg s-2, w=998.9 kg m-3, g=9.81 m s-2, and $\varepsilon$=0°(Mohammadi and Vanclooster 2011).

Then, transforming ri to di and the units to gain Eq. (2) (Eq. (4) in our main manuscript), which is more clear to express the relation between the pore diameter and suction head.

Therefore, we can add the process of transforming Laplace's equation Eq. (1) to Eq. (2) in revised manuscript.

Considering the shape and size of slit spaces were different from the central pore, their suction were calculated using different equations respectively, the suction heads of central pore were calculated using Eq. (2) , while the chemical potentials of slit spaces were calculated using Eq. (3) suggested by (Derjaguin and Churaev 1992), then transforming the units to gain the suction heads.

This aspect have been described in Section 3.1 "Estimating the pore volume fraction" in the main manuscript.

Comment 2: There is also another point, regarding silt-shaped spaces, that in my opinion deserves to be discussed in the paper. To my best understanding, silt-shaped spaces are introduced to consider the water which is bonded to the particles in such a way that the model of the bundle of cylinders fails in describing it. In fact, with such silts dimensions as small as 1 Å are reached. In such a range of dimensions, capillarity is

not anymore the mechanism which bonds water to the soil particles, and other kinds of interactions contribute to the potential energy of water (actually, already for quite larger pore dimensions). So, if equation (4) is still used, this turns out to be an effective, but not physically based, way to obtain water potential.

Reply to comment 2: Recent research have identified the lack of consideration of adsorptive surface forces and liquid films in present theories for flow and transport in unsaturated porous media, which would lead error in corresponding calculation, particularly at low saturation. Nitao and Bear (1996) pointed out that the part of the problem lies in the vague definition of the soil matric potential where capillary and adsorptive forces are lumped together. (Tuller et al. 1999) considered the individual contributions of adsorptive and capillary forces to the matric potential, the liquid-vapor radius of curvature (capillary contribution) and the film thickness were calculation using the same given potential. This simplified method is termed the shifted Young-Laplace (SYL) equation(Tuller and Or 2001). In essence, they made a simplification that the chemical potential, the capillary pressure and the adsorptive pressure were equal.

In our study, a simplification was made that we only take the water in central pore and slit spaces into account, without considering the liquid films coat pore and slit walls, therefore the capillary pressure, as the dominant acting forces, was only considered. Besides, the predicted suction head in our study is lower than 5000 cmH2O, therefore the error resulted from the lack of consideration of adsorptive surface forces were relative small. We will add the discussions corresponding to the silt-shaped spaces in the revised manuscript.

Comment 3: Pag. 5, line 13. The water potential values should be negative.

Reply to comment 3: Indeed, it's true that the critical potential values of the biggest slit spaces should be negative on Page 5, line 13. In order to compare in unified standard, this potential values were transformed into the suction head with unit of cmH2O. It was our oversights that it not be described clearly; hence we will change "critical potential"

as "critical suction head" on Page 5, line 13 in revised manuscript.

Comment 4: Pag. 9, lines 17-19. This statement sounds surprising, if I understand it correctly. The smaller the particles, the larger I expect soil (specific) surface area, as for instance for clay particles. In this respect, the authors should try (where possible), or at least mention the possibility of using measured surface areas rather than estimating it by means of an empirical formula, and discuss how their results could be (positively or negatively) affected.

Reply to comment 4: The surface area (m2) on Page 9, lines 17-19 refer to the total surface area of particle which is positively related to the equivalent particle radius and is different from the specific surface area (m2 g-1).

The direct measurements of the specific surface area were time- and money- consuming and the measuring error would also exist. Furthermore, the measured surface areas for so many samples were difficult to gain for us at present. Therefore calculating the specific surface area using an empirical formula may be the best choice at present. The empirical method used to estimate the specific surface area in Section 3.2 was presented by (Sepaskhah et al. 2010) with an r2 value of 0.88, it proved that this empirical equation have reliable capabilities to use. Although the errors generated inevitably when calculating the specific surface area from the empirical equation, it would enhance analysis uniformity and avoid some error resulted from abnormal measurement.

Comment 5: Pag. 10, line 21. The reference should read "van Genuchten, M. T." instead of "Genuchten, M. T. V.", and the same holds for where such a reference is recalled in the text.

Reply to comment 5: Thank reviewer for pointing out my mistake. "Genuchten, M. T. V." on Page 10, line 21 will change into "van Genuchten, M. T.", and other places where such a reference is used.

References

Haverkamp, R. and Parlange, J.Y. (1986) Predicting the water-retention curve from particle-size distribution: 1. sandy soils without organic matter1. Soil Science 142(6), 325-339.

Mohammadi, M.H. and Vanclooster, M. (2011) Predicting the soil moisture characteristic curve from particle size distribution with a simple conceptual model. Vadose Zone Journal 10(2), 594-602.

Derjaguin, B.V. and Churaev, N.V. (1992) Polymolecular adsorption and capillary condensation in narrow slit pores. Progress in Surface Science 54(2), 157-175.

Nitao, J.J. and Bear, J. (1996) Potentials and Their Role in Transport in Porous Media. Water Resource Research 32(2), 225-250.

Tuller, M., Or, D. and Dudley, L.M. (1999) Adsorption and capillary condensation in porous media: Liquid retention and interfacial configurations in angular pores. Water Resources Research 35(7), 1949–1964.

Tuller, M. and Or, D. (2001) Hydraulic conductivity of variably saturated porous media: Film and corner flow in angular pore space. Water Resources Research 37(5), 1257-1276.

Sepaskhah, A.R., Tabarzad, A. and Fooladmand, H.R. (2010) Physical and empirical models for estimation of specific surface area of soils. Archives of Agronomy & Soil Science 56(3), 325-335.

$$\psi_i = \frac{2\sigma \cos \varepsilon}{r_i g \rho_w} \qquad (1)$$

$$\psi_i = \frac{3000}{d_i} \qquad (2)$$

$$\mu = -2\sigma / (\rho \alpha d) \qquad (3)$$

**Fig. 1.**

---

## Author Comment (AC3) · 3 May 2018

Dear Reviewers and Editor,

We are very grateful to editor and reviewers for their constructive comments and suggestions on our manuscript entitled "Predicting the soil water characteristic curve from the particle size distribution based on a pore space geometry containing slit-shaped spaces". (Manuscript ID: hess-2017-668). We will revise the manuscript according to the comments of reviewers. The main revisions for our manuscript are as follows:

1. Adding more fine-textured soils to validate the proposed model.

2. Comparing proposed model with other improved model (the improved model in Meskini-Vishkaee et al. (2014) may be a good choice) and add the predicted results and discussions.

3. Adding the process of transforming Young-Laplace equation to Eq. (4) in revised manuscript.

4. The discussions corresponding to the silt-shaped spaces will be added in our revised manuscript.

5. Revising the other small points suggested by reviewers. Looking forward to hearing from you.

Best regards,

Dong-hui Cheng

---

## Author Response (AR1)

Dear Editor,

We are very grateful to editor and reviewers for the constructive comments and suggestions on our manuscript (Manuscript ID: hess-2017-668). We have revised the manuscript according to these
5  comments and now submit a point-by-point response, a marked manuscript and a revised manuscript. We hope the revised manuscript would meet with publication requests.

Looking forward to hearing from you

Best regards,

Dong-hui Cheng

35

**A list of all relevant changes made in the manuscript**

5  1. Adding 7 soil samples with more clay content to validate the proposed model.

2.Comparing proposed model with the scaling approach proposed by Meskini-Vishkaee et al. (2014) and also adding the predicted results and discussions.

3. Adding a detailed transformation for Equation (4) in revised manuscript.

4. The discussions corresponding to the slit-shaped spaces have been added in our revised manuscript.

10  5. Revising the other small points suggested by reviewers.

6. Checking and correcting the minor mistakes in the text

35

**A point-by-point response to reviewers and editor**

**Reviewer 1 (F. Meskini-Vishkaee)**

5 **1.** Since the relationship between the PSD and the pore size distribution (PoSD) is a fundamental element when predicting the SWRC from the PSD, first adjective of this study was to compare the estimated PoSD using traditional Method with the measured PoSD. The following comments and responding responses is about PoSD and PSD.

**Comment a:**

10 This step includes i, estimated PoSD from PSD and ii, estimated PoSD from SWCC. The authors have to change subtitle "2) measuring the PoSD" in page 3, line 33 by "2) estimating the PoSD from SWCC".

**Response:**

According to the Reviewer's suggestion, we have changed subtitle "(2) measuring the PoSD" in page 4, line 3 into "(2) estimating the PoSD from SWC".

**Comment b:**

To estimate PoSD from PSD, called the traditional method as Arya model, here a proportionate relationship between pore size and associated particle diameter was used to calculate the equivalent pore diameter (Eqn. 2) because it was easy to use. This simplification may be a part of the estimation error of Arya and Paris (1891)

20 model.

**Response:**

The proportional relationship between the pore size and the associated particle size proposed by Jensen et al (2015) was used to calculate the equivalent pore diameter and the suction head in our manuscript. In order to evaluate the applicability of this method, we calculated the suction head using this proportional relationship and

25 the water content followed the way of A&P model (Arya and Paris, 1981), and predicted the corresponding SWC, moreover compared them with the predicted SWCs using method in MV model (Mohammadi and Vanclooster, 2011) (Figure 1). We concluded that this proportional relationship proposed by Jensen et al. (2015) not only could get a good prediction of suction as the results calculated using the method in Mohammadi and Vanclooster (2011), also was easy to use.

30 It should be noted that this calculation method of suction head was different from that in Arya and Paris (1891) model in which the pore diameters were estimated using the parameter $\alpha$ to scale the pore length and pore volume.

[Figure]

**Figure 1: Measured SWC curves and predicted SWC curves using the methods proposed by Jensen et al (2015) and Mohammadi and Vanclooster (2011) respectively**

**Comment c:**

It is noted that estimation method of PoSD from SWCC is nearly similar to the estimation method of PoSD from PSD proposed by Mohammadi and Vanclooster (2010). Although, since SWCC is influenced both soil texture and structure, if soil organic carbon or clay content would be high, differences between estimated PoSD from SWCC and PSD become more. It must be mentioned that the prediction error of estimated SWCC from PSD is at dry range of SWCC (at high suction heads) that influences by soil texture (especially clay particles). Mohammadi and Meskini-Vishkaee (2012) attribute the methods error to the roughness of soil particles, high surface energy content of clay particles and the simplified pore geometric concepts that does not effectively reflect the pore geometry. It is better that the authors compare estimated PoSD from measured SWCC to estimated PoSD from PSD using similar method (use Mohammadi and vanclooster method as traditional method). Therefore, I think that these calculations have to add to this part of manuscript.

**Response:**

We agree with the reviewer's viewpoint that the error of predicted SWC from PSD in dry range (at high suction heads) that is influenced by soil texture (especially clay particles).

As mentioned in the comments, the estimation method of SWC from PSD is nearly similar to the estimation method proposed by Mohammadi and Vanclooster (2011). Mohammadi and Vanclooster (2011) calculated the suction head by assuming a linear relationship between the suction head and packing state, and the packing state is estimated from particle and bulk densities. Their suction head calculation method is given by

$$\psi_i = \frac{0.543 \times 10^{-4}}{R_i} \xi$$

(1)

where $\xi$ is a coefficient depending on the state of soil particles packing; $R_i$ is the particle radius. When calculating PoSD from SWC, a critical step is to estimate the pore size from the suction, but the calculation method of suction head in Mohammadi and Vanclooster (2011) (Eq.(1)) is the function related the suction head and the particle radius, not the relation of the suction head and the pore size. Therefore it could not be used for calculating PoSD from SWC.

**2.** Tuller et al. (1999) and Or and Tuller (1999) proposed including the water films coating the pore walls and water in angular spaces of pores, in calculations of soil water content. Despite great scientific interest, the proposed approach for the derivation of SMC by Or and Tuller (1999) motivated by bundle of cylindrical tubes limitations, usually fails to describe experimental data in the intermediate soil water content range because of the low flexibility of the gamma distribution function used to characterize the PoSD (Lebeau and Konrad, 2010). In addition, the model is mathematically complex and furthermore needs specific surface area parameter which measurements and estimations are often quite variable (Carter et al., 1986). The following two comments and responses are related to the pore geometry model and its parameters.

**Comment a:**

The authors use pore geometry containing slit-shaped spaces proposed by Or and Tuller (1999), But they assumed that circle-shaped central pore connected to two slit-shaped spaces. Moreover, the estimated PoSD data were fitted using a modified logistic growth model (Eqn. 5).

**Response:**

We go along with the reviver's comments above. The pore geometry containing slit-shaped spaces proposed by Or and Tuller (1999) is regarded as a more realistic description for natural pore spaces. We used closely similar pore model which containing circle-shaped central pore connected to two slit-shaped spaces to predict SWC and could get the good fit with the measured SWC.

**Comment b:**

Specific surface area (SSA) is a requirement parameter to obtain the values of $\alpha$ and $\beta$. The authors used a power equation with two fitting parameters (Eqn. 10) to estimate SSA proposed by Sepaskhah et al. (2010). Sepaskhah et al. (2010) used twenty soil samples from a depth of 0–30 cm were collected from different locations in Fars

province, in the south of Iran to calibrate the power equation. In addition, a different set of data was used to validate the calibrated model. Their results indicated that in the range of around 20 up to 200 $m^2 g^{-1}$ the values of measured SSA were in quite a good agreement, while for SSA greater than 200 $m^2 g^{-1}$, the deviations increase distinctly. Moreover, Tuller and Or (2005) stated that the psychrometric approach for SSA determination should provide reliable values for natural soils with hydratable surface areas below 200 $m^2 g^{-1}$. They recommend using SWCC values for -10 MPa and lower (drier) with an effective Hamaker constant of $-6 \times 10^{-20}$ J to predict SSA values. So, there are some ambiguities here,

i. As respects higher SSA is related to finer texture soils that usually have underestimation problem of estimated SWCC from PSD, Indeed, I think use power model to estimate SSA cannot be useful to improve estimated SWCC in fine-textured soils. Page 9, line 4: the authors declared that "for the coarse-textured soil, the water content and prediction error of the SWCC changed relatively little for the same degree of change of the SSA". This is completely expected because not only there is not serious problem to estimate SWCC from PSD in coarse-textured soils, but also the value of estimated SSA using power equation is below 200 $m^2 g^{-1}$ for coarse-textured soils.

ii. Is there any SSA measurement? Were the fitting parameters of power model controlled?

**Response:**

(1) The response to the comment on the SSA estimation error

As reviewer mentioned above that the power function used to estimate the SSA are in quite a good agreement with the measured SSA when the values of measured SSA are in the range of around 20 up to 200 $m^2 g^{-1}$, while for measured SSA greater than 200 $m^2 g^{-1}$, the deviations increase distinctly (Sepaskhah et al., 2010). Moreover, their study showed that the power function with an $r^2$ value of 0.88 is superior to the physical model and the multivariate pedo-transfer function for the estimation of SSA.

In our method, the values of parameter $\alpha$ and $\beta$ were firstly figured out using SSA and the measured SWC, and then these parameters were used for predicting SWC as input parameters. For the predicted SWCs of fine-textured soils which calculated from the parameter $\alpha$ and $\beta$, the errors from estimated SSA, to some extend, could been offset by the parameter $\alpha$ and $\beta$. Besides, the parameter $\alpha$ and $\beta$ were main used to estimate the volume fraction of the slit-shaped spaces, thus the estimation accurate of SSA influence the estimation of the volume fraction of the slit-shaped spaces, consequently the degree of improvement of predicted SWC. Overall there are always different levels of improvement comparing with the SWC predicted by the traditional method for all samples. Certainly, more effort should be directed to a more accurate method of SSA estimation.

We have added the discussions about the effect of the power equation to the SSA estimation in revised manuscript.

(2) The response to comments on the effect of SSA to the coarse-textured soil

As pointed out by reviewer, the factors that the water content and the prediction error of the SWC changed relatively little under the same proportional change of the SSA for coarse-textured soils include two aspects. We have enriched some discussions in Section 4.3.2 in our revised manuscript.

Meanwhile, although the absolute errors of predicted water content to measured water content using traditional models for fine-textured is higher than that for coarse-textured soil. However the relative errors of both fine-textured and coarse-textured soils cannot be ignored.

(3) The response to the comments on the SSA measurement

We regret that we do not conduct SSA measurement. The power equation employed to predict SSA in our manuscript is an empirical equation, and the parameters values in our manuscript cited Sepaskhah et al. (2010).

**3.** At the first step, the estimated PoSDs of 48 soil samples using SWRC were compared with the PoSDs calculated using PSD to identify the origins of the errors and their effects on the accuracy of the SWC and to calibrate the proposed model. Subsequently, 22 soil samples were also selected from UNSODA database to validate the model. The following three comments and responses are related to the data sets of the soil samples

**Comment a:**

Please provide a Table involved some properties of selected samples for both calibration and validation data sets (e.g. max, min and average of clay content, organic matter, bulk density and……for each soil textural class).

**Response:**

The detailed information for both validation and calibration data sets are presented in Table 1and Table 2, respectively.

**Table 1: Basic soil properties of 29 samples for the model validation.**

| Soil texture | Number of soil | | Clay (%) | Sand (%) | $\rho_b$ (g m$^{-3}$) |
|---|---|---|---|---|---|
| | 5 | Min | 43 | 5.4 | 1.1 |
| Clay | | Max | 57 | 32 | 1.5 |
| | | Average | 51 | 14.4 | 1.31 |
| Loamy | 2 | Min | 16.5 | 47.9 | 1.41 |
| | | Max | 29.2 | 43.6 | 1.45 |

| Soil texture | Number of soil | | Clay (%) | Sand (%) | $\rho_b$ (g m$^{-3}$) |
|---|---|---|---|---|---|
| Loamy sand | 5 | Min | 1.7 | 75.5 | 1.37 |
| | | Max | 7.3 | 85.2 | 1.59 |
| | | Average | 4.9 | 81 | 1.46 |
| Sand | 5 | Min | 1.1 | 90.1 | 1.46 |
| | | Max | 4.4 | 97.5 | 1.58 |
| | | Average | 2.3 | 93.4 | 1.53 |
| Sandy loam | 2 | Min | 11.4 | 56.8 | 1.44 |
| | | Max | 12.6 | 65.7 | 1.46 |
| Sandy clay loam | 6 | Min | 9.8 | 28 | 1.21 |
| | | Max | 30.7 | 69.7 | 1.53 |
| | | Average | 22.8 | 43.2 | 1.45 |
| Clay loam | 4 | Min | 33.4 | 20.4 | 1.07 |
| | | Max | 37.5 | 34.7 | 1.58 |
| | | Average | 35.1 | 24.8 | 1.27 |

**Table 2: Basic soil properties of 48 samples for the model calibration.**

| Soil texture | Number of soil | | Clay (%) | Sand (%) | $\rho_b$ (g m$^{-3}$) |
|---|---|---|---|---|---|
| Clay | 9 | Min | 41.5 | 6.1 | 1.08 |
| | | Max | 58.2 | 36 | 1.64 |
| | | Average | 50.2 | 14.1 | 1.29 |
| Loam | 4 | Min | 14 | 42 | 1.36 |
| | | Max | 23 | 67 | 1.63 |
| | | Average | 17.3 | 50.5 | 1.46 |
| Loamy sand | 10 | Min | 3 | 76.2 | 1.32 |
| | | Max | 10.4 | 89.4 | 1.6 |
| | | Average | 6.1 | 83.2 | 1.46 |
| Sand | 16 | Min | 0.7 | 89.6 | 1.41 |
| | | Max | 4.6 | 98.9 | 1.7 |
| | | Average | 2.5 | 93.4 | 1.55 |
| Sandy clay loam | 1 | | 2.7 | 62.5 | 1.7 |
| Sandy loam | 3 | Min | 10.5 | 64.9 | 1.27 |
| | | Max | 19.4 | 76.3 | 1.7 |

| | | Average | 15 | 68.8 | 1.50 |
|---|---|---|---|---|---|
| Silt loam | 5 | Min | 10.5 | 21 | 1.49 |
| | | Max | 15.7 | 34.8 | 1.56 |
| | | Average | 12.6 | 26.5 | 1.52 |

**Comment b:**

About validation data set, Textural distribution of the 22 soil samples is shown in both Figure 5 and Table 3. This duplication is not necessary.

**Response:**

Figure 5 in manuscript has been deleted in order to avoid repetition.

**Comment c:**

As regards the most prediction error of traditional models is often related to soils with good structure or high clay content. Therefore, the authors have to use more fine-textured soils to validate their proposed model. In validation data set, only 4 soil samples had clay texture and more than 60 % of soil samples are coarse-textured soils. Please add more soil samples with higher clay content and organic matter to the validation data set.

**Response:**

We have added 7 soil samples with clay content larger than 20% in order to fully validate the predicted model. The added soil samples were summarized in Table 3.

**Table 3: Codes and textural classes of the added 7 soils selected from UNSODA**

| UNSODA codes | Textual class |
|---|---|
| 1320 | Clay |
| 1102, 2341 | Sandy clay loam |
| 3031, 3032, 1372, 1362 | Clay loam |

We have predicted the SWCs using the improved method, the scaling approach and the traditional method for the added soil samples respectively, and their predicted results and discussion have been added in revised manuscript.

4. The following comment is related to the calibration and validation of proposed model.

Comment:

In page 8, line 19-21: the authors stated that "These improvements are mainly attributed to the pore model containing slit-shaped spaces, demonstrating that this pore model is better for predicting the SWC from the PSD than the concept of a bundle of cylindrical tubes". This simplification (concept of a bundle of cylindrical tubes) is introduces as major source of error in the SWCC predictor models using PSD. After that, some studies have

attempted to improve the water content calculation approach by attributing model errors to both a simplified pore geometry and an incomplete desorption of residual water in the soil pore within the high matric suction head range. Therefore, I think the authors have to compare proposed model to other models except Arya and Paris (1981), such as Mohammadi and Meskini-Vishkaee (2012) or Meskini-Vishkaee et al. (2014) or other models. The comparison between the performance of these models and parameter needs can be more helpful. Please expand discussion part and state the result of proposed model for both data sets (calibration and validation) in more detail。

**Response:**

We agree that quality of our manuscript will improve if the performances comparison between the proposed model and other improved models are added. Thus, we have compared our model with a scaling approach proposed by Meskini-Vishkaee et al. (2014) and added the predicted results and discussions in Section 4.2 in revised manuscript. (The estimation of SWCs for validation data are listed in Fig.S3 in the supporting information).

The results illustrated that the improved method here applied well to a wide range of soils, while the scaling approach performed better for fine- and medium-textured soils. The $E_{RMS}$ values range from 0.017 to 0.054 for the improved method (with an average of 0.028), from 0.026 to 0.060 for the scaling approach (with an average of 0.037) and from 0.040 to 0.106 for the traditional method (with an average of 0.061). Among the three methods mentioned above, the improved method provided the best predictions and the traditional method performed worst.

**Reviewer 2**

**Comment 1:** Detailed comments Equation (4) should be rewritten in a more general way, regardless of the units adopted for the water potential. In this regard, it seems that this equation is used to link pore dimension to water potential, even in the silt-shaped space between pores. This aspect should be better clarified, as the dimension of the silts are proportional to the pore diameter, so it is not clear what is the diameter introduced in equation (4) to obtain the corresponding potential.

**Response to comment 1:**

(1) The transformation process of Equation (4) in the main manuscript

Equation (4) in our main manuscript was gained by substituting known parameters into Laplace's equation (Eq. (2))(Haverkamp et al., 1986), in which $\sigma=7.275\times10^{-2}$ kg s$^{-2}$, $\rho_w=998.9$ kg m$^{-3}$, $g=9.81$ m s$^{-2}$, and $\varepsilon=0°$ (Mohammadi and Vanclooster, 2011).

$$\psi_i = \frac{2\sigma\cos\varepsilon}{r_i g \rho_w} \tag{2}$$

Then, transforming $r_i$ to $d_i$ and the units to gain Eq. (3) (Eq. (4) in our main manuscript), which is more clear to express the relation between the pore diameter and suction head.

$$\psi_i = \frac{3000}{d_i} \tag{3}$$

We have added the transformation process above in our revised manuscript.

(2) The suction head calculation for slit-shaped spaces

Because the shape and size of slit spaces were different from the central pore, their suction heads were calculated using different equations respectively. The suction heads of central pore were calculated using Eq. (3) , while the chemical potentials of slit spaces were calculated using Eq. (4) suggested by Derjaguin and Churaev (1992) and then transforming the units to gain the suction heads.

$$\mu = -2\sigma / (\rho\alpha d) \tag{4}$$

Where, $\alpha$ is the scaling parameter of the slit width.

This aspect have been rewrote in Section 3.1 "Estimating the pore volume fraction" in the revised manuscript.

**Comment 2:** There is also another point, regarding silt-shaped spaces, that in my opinion deserves to be discussed in the paper. To my best understanding, silt-shaped spaces are introduced to consider the water which

is bonded to the particles in such a way that the model of the bundle of cylinders fails in describing it. In fact, with such silts dimensions as small as 1 Å are reached. In such a range of dimensions, capillarity is not anymore the mechanism which bonds water to the soil particles, and other kinds of interactions contribute to the potential energy of water (actually, already for quite larger pore dimensions). So, if equation (4) is still used, this turns out to be an effective, but not physically based, way to obtain water potential.

**Response to comment 2:**

Nitao and Bear (1996) pointed out that the vague definition of the soil matric potential where capillary and adsorptive forces are lumped together. During drainage, when considering the capillary forces only, the drainage potential in slit-shaped pore is given as Eq.(7) (Derjaguin and Churaev, 1992) in revised manuscript, while the applicability of this formula is limited by condition the width of the slit. When passing over to thin slits, a correction will have to be introduced, taking into account the effect of adsorption force at the slit surfaces. Tuller and Or (2001) defined a critical slit spacing ($\alpha d^*$) by Eq.(16) in revised manuscript that would classify slit sizes responding to capillary drainage and adsorption dominated drainage. In case of slit spacing greater than $\alpha d^*$, the capillary-based slit snap-off would be applied. The value of $\alpha d^*$ is 0.591 nm, it means that for slit spacing greater than 0.591 nm, the Eq.(7) could be applied to calculate the drainage potential in slit-shaped pore in revised manuscript.

Besides, a simplification was made in our study that we only take the water in central pore and slit spaces into account, without considering the liquid films coat pore and slit walls; therefore the capillary pressure, as the dominant acting forces, was only considered. Furthermore because the predicted suction head in our study is lower than 5000 $cmH_2O$, the error resulted from the lack of consideration of adsorptive surface forces were relative small. We have added a chapter to discuss the property of the slit-shaped spaces in the revised manuscript.

**Comment 3:** Pag. 5, line 13. The water potential values should be negative.

**Response to comment 3:**

Indeed, it's true that the critical potential values of the biggest slit spaces should be negative on Page 5, line 13. In order to compare in unified standard, this potential values were transformed into the suction head with unit of $cmH_2O$. It was our oversights that it not be described clearly; hence we have changed "critical potential" as "critical suction head" on Page 5, line 12 in revised manuscript.

**Comment 4:** Pag. 9, lines 17-19. This statement sounds surprising, if I understand it correctly. The smaller the particles, the larger I expect soil (specific) surface area, as for instance for clay particles. In this respect, the authors should try (where possible), or at least mention the possibility of using measured surface areas rather than estimating it by means of an empirical formula, and discuss how their results could be (positively or negatively) affected.

**Response to comment 4:**

The surface area ($m^2$) on Page 9, lines 17-19 refer to the surface area of particle which is positively related to the equivalent particle radius and is different from the specific surface area ($m^2 g^{-1}$) which is the total surface area of a material per unit of mass.

As reviewer mentioned that the deviations will generate when estimate the SSA using the power function. Moreover, their study showed that the $r^2$ between the SSA predicted by the power function and the measured SSA is 0.88, it proved that this empirical equation have reliable capabilities to use.

Furthermore, the measured SSAs for so many samples were difficult for us at present. Therefore calculating the specific surface area using an empirical formula may be the best choice.

**Comment 5:** Pag. 10, line 21. The reference should read "van Genuchten, M. T." instead of "Genuchten, M. T. V.", and the same holds for where such a reference is recalled in the text.

**Response to comment 5:** Thank reviewer for pointing out our mistake. "Genuchten, M. T. V." on Page 10, line 21 have changed into "van Genuchten, M. T.".

[revised manuscript text omitted]

---

## Referee Report (RR1)

**F. Meskini-Vishkaee**

The authors propose a modified approach to estimate the soil moisture characteristic curve (SMC) from the measurement of the particle size distribution (PSD) and specific surface area (SSA). The authors used an empirical power equation with two fitting parameters to estimate SSA proposed by Sepaskhah et al. (2010). Subsequently, estimated SSA is used to estimate two other fitting parameters ($\alpha$ and $\beta$), and at the end estimated $\alpha$ and $\beta$ are used to estimate the slit pore volume fraction. In spite of adding seven soil samples with more clay content in the validation dataset and Comparing proposed model with the scaling approach proposed by Meskini-Vishkaee et al. (2014), result and discussion part of manuscript must be extended.

The minimal requirements for possible publication mandate the following revisions:

i: In response to reviewer's comment "Specific surface area (SSA) is a required parameter to obtain the values of $\alpha$ and $\beta$. The authors used a power equation with two fitting parameters (Eqn. 10) to estimate SSA proposed by Sepaskhah et al. (2010). Sepaskhah et al. (2010) used twenty soil samples from a depth of 0–30 cm were collected from different locations in Fars province, in the south of Iran to calibrate the power equation. In addition, a different set of data was used to validate the calibrated model. Their results indicated that in the range of around 20 up to 200 m^2 g^-1 the values of measured SSA were in quite a good agreement, while for SSA greater than 200 m^2 g^-1, the deviations increase distinctly. As respects higher SSA is related to finer texture soils that usually have underestimation problem of estimated SWCC from PSD, Indeed, I think use power model to estimate SSA cannot be useful to improve estimated SWCC in fine-textured soils. ", the authors mentioned that in proposed method, the values of parameter $\alpha$ and $\beta$ were firstly figured out using SSA and the measured SWC, and then these parameters were used for predicting SWC as input parameters. For the predicted SWCs of fine-textured soils which calculated from the parameter $\alpha$ and $\beta$, the errors from estimated SSA, to some extend, could been offset by the parameter $\alpha$ and $\beta$. Besides, the parameter $\alpha$ and $\beta$ were main used to estimate the volume fraction of the slit-shaped spaces, thus the estimation accuracy of SSA influence the estimation of the volume fraction of the slit-shaped spaces, consequently the degree of improvement of predicted SWC.

The main objective to estimate SMC from PSD is to have SMC data when this data is not available. However, as mentioned above, SMC data is needed to develop proposed model for the accurate estimation especially for the fine-textured soils.

ii: regardless good performance of proposed model, SSA is used to develop model and SSA measurement is very difficult, thus it must be estimated. The proposed model is an estimation method that needs to estimate its input parameters. It is a disadvantage of proposed model.

The authors noted that the results illustrated that the improved method here applied well to a wide range of soils, while the scaling approach performed better for fine- and medium-textured soils. The validation results illustrate that the SMC predicted using the proposed method

provided the best predictions of the SMC, closely followed by the scaling approach, and the traditional method performed worst. Did the authors do any statistical analysis between the performances of three models? Is there any significant difference between three models? The authors could perform a paired T-test analysis between proposed and Meskini-Vishkaee models for mean model performance in different soil textural classes.

Even if there was a significant difference between two desired models, the proposed model is a complex model with some input parameters that is need to estimate from some difficult properties such as SSA. It is correct that the assumption of pore space geometry containing slit-shaped spaces may be affected on the accuracy of the estimation, but on the other hand, this assumption could be increased the model inputs and complexity.

I think that the authors have to add some more discussion to explain the advantage and disadvantages of proposed model. Moreover, performing statistical analysis between model performances is necessary and must be added to the manuscript text.

Minor revision:
The authors provided detailed information of both validation and calibration data sets in two table in response to reviewers. I think these tables have to add in main text of manuscript. Note: offering of the statistical criteria for each soil textural classes is not necessary. Presenting of the mean, max and min of all data for both dataset is enough.

---

## Author Response (AR2)

Dear Editor,

We would like to express our great appreciation to editor and reviewers for their constructive comments on our manuscript (Manuscript ID: hess-2017-668). We have revised the manuscript according to these comments and now submit a point-by-point response, a marked manuscript and a revised manuscript. We hope the revised manuscript would meet with publication requests.

In addition, the cost of English Language Editing for this manuscript was also was supported by the National Natural Science Foundation of China (41472220) and we hope to add it in Acknowledgement. We would be greatly appreciated if you could allow our request.

Looking forward to hearing from you

Best regards,

Dong-hui Cheng

**A list of all relevant changes made in the manuscript**

1. Performing T-test analysis between the performances of the three methods, and adding the corresponding results in revised manuscript.

5  2. Adding the advantage and disadvantages of proposed method in Section 4.2.

3. Adding two tables of detailed information for both validation and calibration data sets in revised manuscript.

4. Rewritten Section 3.1

5. Revising Section 4.3.1

6. Editing the English language of the whole manuscript

10  7. Revising other small points suggested by reviewers.

30

35

**A point-by-point response to reviewers and editor**

**Reviewer 1 (F. Meskini-Vishkaee)**

5   **1.**In response to reviewer's comment "Specific surface area (SSA) is a required parameter to obtain the values of α and β. The authors used a power equation with two fitting parameters (Eqn. 10) to estimate SSA proposed by Sepaskhah et al. (2010). Sepaskhah et al. (2010) used twenty soil samples from a depth of 0–30 cm were collected from different locations in Fars province, in the south of Iran to calibrate the power equation. In addition, a different set of data was used to validate the calibrated model. Their results indicated that in the range of around 20 up to 200 $m^2 g^{-1}$ the

10  values of measured SSA were in quite a good agreement, while for SSA greater than 200 $m^2 g^{-1}$, the deviations increase distinctly. As respects higher SSA is related to finer texture soils that usually have underestimation problem of estimated SWCC from PSD, Indeed, I think use power model to estimate SSA cannot be useful to improve estimated SWCC in fine-textured soils. ", the authors mentioned that in proposed method, the values of parameter α and β were firstly figured out using SSA and the measured SWC, and then these parameters were used for predicting SWC as

15  input parameters. For the predicted SWCs of fine-textured soils which calculated from the parameter α and β, the errors from estimated SSA, to some extent, could been offset by the parameter α and β. Besides, the parameter α and β were main used to estimate the volume fraction of the slit-shaped spaces, thus the estimation accuracy of SSA influence the estimation of the volume fraction of the slit-shaped spaces, consequently the degree of improvement of predicted SWC.

20   **Comment a:**

The main objective to estimate SMC from PSD is to have SMC data when this data is not available. However, as mentioned above, SMC data is needed to develop proposed model for the accurate estimation especially for the fine-textured soils.

**Response:**

25  The SWRC data is not needed until developing proposed model. When predicting the SWRC, required input data include the PSD, the measured water content and the bulk and particle densities.

**2.** Regardless good performance of proposed model, SSA is used to develop model and SSA measurement is very difficult, thus it must be estimated. The proposed model is an estimation method that needs to estimate its input

30  parameters. It is a disadvantage of proposed model.

**Comment a:**

The authors noted that the results illustrated that the improved method here applied well to a wide range of soils, while the scaling approach performed better for fine- and medium-textured soils. The validation results illustrate that the

SMC predicted using the proposed method provided the best predictions of the SMC, closely followed by the scaling approach, and the traditional method performed worst. Did the authors do any statistical analysis between the performances of three models? Is there any significant difference between three models? The authors could perform a paired T-test analysis between proposed and Meskini-Vishkaee models for mean model performance in different soil textural classes.

**Response:**

We have performed a T-test analysis between the performances of three methods. The results showed that there is a significant difference between performance of improved method and traditional method (p=0.001). Only for sand samples, the performance of improved method and scaling approach have significant statistics difference (p=0.01).This content have been added in Section 4.2.

**Comment b:**

Even if there was a significant difference between two desired models, the proposed model is a complex model with some input parameters that is need to estimate from some difficult properties such as SSA. It is correct that the assumption of pore space geometry containing slit-shaped spaces may be affected on the accuracy of the estimation, but on the other hand, this assumption could be increased the model inputs and complexity.

**Response:**

As the reviewer mentioned that the principle of proposed model is complex, but if you have understood its calculation procedure, it would be easy to predict the SWRC of multiple samples in Excel or other software.

**Comment c:**

I think that the authors have to add some more discussion to explain the advantage and disadvantages of proposed model. Moreover, performing statistical analysis between model performances is necessary and must be added to the manuscript text.

**Response:**

We have added the statistical analysis results between model performances in Section 4.2. Meanwhile, we have added the advantage and disadvantages of proposed model in this section.

3. The authors provided detailed information of both validation and calibration data sets in two tables in response to reviewers. I think these tables have to add in main text of manuscript. Note: offering of the statistical criteria for each soil textural classes is not necessary. Presenting of the mean, max and min of all data for both dataset is enough.

**Response:**

We go along with the suggestion of reviver above, and we have added two tables of detailed information for both validation and calibration data sets in revised manuscript.

30

**Reviewer 2**

**Comment 1:** Throughout the entire manuscript, I strongly suggest using "soil water retention curve", rather than "soil water characteristic curve", as it is by far the more established (and clear) name for the curve you are estimating (indeed, usually two soil water characteristic curves are considered: water retention curve and hydraulic conductivity curve). Consistently, I would adopt the acronym SWRC (or simply WRC) instead of SWC.

**Response:**

We agree with the reviewer's viewpoint above, and we have adopted "soil water retention curve" instead of "soil water characteristic curve" in the revised manuscript.

**Comment 2:** Section 3.1

This modified section now gives more information about how water potential in silt shaped voids has been calculated. However, this section should be rewritten and better organized:

**Response:**

We go along with the comments of reviver above. This section has been rewritten in the revised manuscript.

**Comment 3:** Page 5, line 20: the values of the parameters of Table 2 are here discussed before introducing Table 2 and the way such parameters were estimated. Section 3.1 should contain only general (theoretical) arguments, and the eventual confirmation given by the estimated parameters should be discussed afterwards.

**Response:**

We agree with the reviewer's viewpoint above, we have moved the eventual confirmation given by the estimated parameters to Section 4.3.1.

**Comment 4:** Page 5, equation (7): I don't see the necessity of writing this equation, as it is exactly the same as equation (4) (cos $\varepsilon$ = 1), written in terms of water potential rather than in terms of suction head. You should simplify your discussion by simply stating that you are using exactly the same equation in circular voids and in slits.

**Response:**

Although the capillary theory are commonly used in equation (4) and equation (7) which calculated the suction head in central pore and slit-shaped spaces, the equivalent pore radius are different in these two equations. Consequently, it's more understandable to write equation (4) and equation (7) in manuscript. We revised the corresponding discussion but retained the original content.

**Comment 5:** Page 5, lines 24-26: you state that 6502 is smaller than 5000, while it is obviously not so. This is again the issue of the intrinsic negative values of water potential that I already raised in my previous report. You should decide if you want to refer to suction head (as you write in your answer to my previous comment), or to water potential (as in equation 7), and then stick consistently to this choice throughout the entire manuscript.

5 **Response:**

Thank reviewer for pointing out my mistake. The suction head will be used throughout the entire manuscript, the false statement on Page 5, lines 24-26 has been revised,

**Comment 6:** Page 5, lines 25-26: I don't agree with this statement. The values of suction head do not demonstrate 10 anything about the dimensions of the voids where the meniscus is supposed to be located. They are, instead, a consequence of the dimensions, which are in turn a consequence of the estimated values of the parameters $\alpha$ and $\beta$, that you are discussing here, before explaining how they were estimated and which results you obtained (see my previous comment in this respect).

**Response:**

15 On the basis of capillary theory, suction head can be associated with pore radius with the capillary equation (Ding et al., 2016). Besides, Jayakody et al. (2014) calculated the pore size using the capillary theory. Consequently, the pore size can be estimated using suction head under the assumption of considering the capillary forces only. We have revised the statement to make it clear and this aspect have been moved to Section 4.3.1.

20 **Comment 7:** Page 5, line 27: the word "included", instead of "contained", would be more appropriate and would make the concept clearer to the reader. My concerns about soil surface area and soil specific surface area have been addressed in section 4.3.2. Few minor issues in the newly added parts:

**Response:**

Considering the reviewer's suggestion above, the word "included" have been used in the revised manuscript.

**Comment 8:** Page 10, line 10: The meaning of the sentence "This effect may contribute to the lower SSA value for this texture than the fine-textured soil" is obscure. Please, reformulate it, or delete it.

**Response:**

30 According to the Reviewer's suggestion, we have deleted the sentence "This effect may contribute to the lower SSA value for this texture than the fine-textured soil".

**Comment 9:** Page 10, line 13: What does it mean "soil media data"?

**Response:** "soil media data" is a loose phrase, we have adopted "soil properties" in the revised manuscript.

**Comment 10:** Page 10, lines 19-21: The syntax of the two sentences, from "Overall" till the end of the section, is wrong.

**Response:** We have revised the sentences as "Therefore, more effort should be placed toward developing a more accurate transformation from the soil physical properties to $S_{SA}$ to further improve the prediction of the SWRC."

**Comment 11:** Section 4.3.3 is called "Physical meanings of the parameters", while it actually gives no physical interpretation of the obtained values. Just a comparison with the results of the conceptually similar model by Or and Tuller (1999) is proposed. I also observe that the adopted model of the pore geometry is just conceptual (and not physical at all), so I would refrain from claiming that the obtained parameter values have any physical meaning, and I would use another name for this section.

**Response:**

We agree with the reviewer's viewpoint above, the name of Section 4.3.3 has revised as "The slit-shaped spaces and the $S_{SA}$ at the sample scale".

**References**

Ding, D., Zhao, Y., Feng, H., Peng, X., and Si, B.: Using the double-exponential water retention equation to determine how soil pore-size distribution is linked to soil texture, Soil & Tillage Research, 156, 119-130, 2016.

[revised manuscript text omitted]

---

## Author Response (AR3)

Dear Editor,

We would like to express our great appreciation to editor and reviewers for their constructive comments on our manuscript (Manuscript ID: hess-2017-668). We have corrected some small mistakes in the manuscript and now
5  submit a marked manuscript and a revised manuscript.

Looking forward to hearing from you

Best regards,
10
Dong-hui Cheng

[revised manuscript text omitted]

